# CROSS-VIEW LEWIS WEIGHT FUSION EMPOWERING EXEMPLAR REPLAY FOR FEDERATED CLASS-INCREMENTAL LEARNING

## ABSTRACT

Federated Class-Incremental Learning (FCIL) aims to continually expand a model's recognition capacity in a distributed environment, enabling it to learn new classes while retaining knowledge of previously seen ones. Exemplar replay has emerged as a promising strategy owing to its simplicity and effectiveness. Existing methods either select exemplars based on local dynamics or construct global feature spaces to identify representative samples. However, they face inherent challenges in striking a balance between effectiveness and privacy. To address this issue, this paper proposes a Cross-view Lewis weIght Fusion method for exemplar replay in FCIL, termed `CLIF`, which fuses multi-view importance scores to guide representative sample selection under federated settings. Specifically, `CLIF` consists of two main modules: 1) the cross-view Lewis weight fusion module computes and integrates Lewis weights from multiple feature perspectives to achieve consistent importance estimation, ensuring that the selected samples better reflect the global data distribution and thus enhancing the representativeness of the replay subset. Building on this, 2) the frequency-based weighted training module adjusts the loss contribution of each sample according to its selection frequency across views, which emphasizes the contribution of critical samples. Moreover, we provide a theoretical analysis to guarantee the soundness and effectiveness of `CLIF`. Extensive experiments on three datasets demonstrate that our method consistently improves baselines by 1%–6%, supporting the above claims.

## 1 INTRODUCTION

Federated learning (FL) enables distributed collaboration to train a shared model using diverse data sources without centralizing them (Qi et al., 2023b; Hu et al., 2024). This follows the principle of "moving models instead of data", which enhances data privacy protection (Banabilah et al., 2022; Liao et al., 2024). However, data heterogeneity among participants inevitably induces divergence in client model update directions, undermining the effectiveness of global aggregation (McMahan et al., 2017; Wang et al., 2023). More critically, in real-world deployments, edge devices are required to handle continuously emerging classes with evolving distributions, requiring models to learn new classes while avoiding forgetting old ones, which has given rise to research on Federated Class-Incremental Learning (FCIL) (Dong et al., 2022; Nori et al., 2025). These scenarios integrate challenges such as data heterogeneity, storage limitations, and continual task evolution, placing higher demands on the scalability of FL systems (Chen et al., 2024; Babakniya et al., 2023).

To tackle these issues, two main strategies have been commonly adopted in existing studies: generative replay-based and exemplar replay-based approaches. The former methods typically train a generative model to synthesize pseudo-samples of previously learned classes, which are then replayed together with new data to alleviate catastrophic forgetting (Wang et al., 2024; Nguyen et al., 2024; Zhang et al., 2023). Without relying on the storage of large-scale raw samples, they achieve stronger privacy guarantees while mitigating memory usage (Yu et al., 2024; Babakniya et al., 2023). However, their effectiveness highly depends on the quality of the generated samples, and the training of generative models often introduces computational and communication overhead, which limits their scalability on resource-constrained edge devices (Tran et al., 2024). In contrast, the latter methods maintain a small memory buffer to store representative exemplars from previous tasks and replay

Figure 1: The proposed cross-view replay (Fig. (b)) fuses Lewis weights from multiple views to select representative samples, overcoming the single-view bias of local replay (Fig. (a)). As shown in Fig. (c), it achieves lower cross-space reconstruction error (CSRE), while avoiding the privacy risks and communication overhead of global-view feature sharing (Fig. (d)).

them alongside new data (Li et al., 2024a). They either select exemplars according to local training dynamics (Li et al., 2025) or construct a global feature matrix to identify representative samples (Qi et al., 2025). This strategy is simple, efficient, and has shown strong empirical performance. Nevertheless, local-view strategies neglect cross-client collaborative features, diminishing the effectiveness of global aggregation, while global-view methods address this limitation at the cost of potential privacy risks, as they may leak feature statistics and spectral information (even without directly sharing raw features, as in FedCBDR (Qi et al., 2025)), as shown in Figure 1.

To address this problem, this paper proposes CLIF, a Cross-views Lewis weIght Fusion framework designed for exemplar replay in FCIL. The key idea is to integrate importance scores derived from multiple feature perspectives, achieving a more consistent and robust evaluation of sample representativeness across heterogeneous clients. Specifically, CLIF is composed of two key components. The first, the cross-view Lewis weight fusion module, constructs multiple feature views by combining different subsets of client models to extract representations. By computing Lewis scores under these diverse client-model combinations and integrating them into a unified importance estimation, the module ensures that the selected exemplars capture a broader spectrum of global knowledge, improving the representativeness of the replay buffer. Subsequently, the second component, the frequency-aware weighted training module, strategically modulates each exemplar's influence on the optimization objective in proportion to its cross-view selection frequency, which amplifies the role of consistently critical samples and strengthens the model's resilience to forgetting. In addition, we provide a theoretical analysis that formally demonstrates the effectiveness of CLIF, offering solid guarantees of its reliability and further reinforcing the soundness of its design.

Extensive experiments were performed on three datasets across different heterogeneity settings and client numbers, involving performance evaluation, ablation studies, hyperparameter analysis, and case studies. The results show that the replayed samples identified by CLIF effectively preserve the underlying geometric structure to the greatest extent. Moreover, CLIF outperforms the strongest of eight state-of-the-art baselines, yielding a 1%–6% increase in Top-1 accuracy.

## 2 RELATED WORK

### 2.1 GENERATIVE REPLAY-BASED METHODS

In FCIL, generative replay constitutes an important line of research for alleviating catastrophic forgetting. Its core principle is to leverage generative models to synthesize samples or features of previously learned classes, which are then rehearsed alongside current-task data (Rong et al., 2025; Wang et al., 2024; Qi et al., 2023a; Mei et al., 2024; Churamani et al., 2024; Zhang et al., 2023). This paradigm preserves prior knowledge and mitigates forgetting. Within the federated setting, generative replay further offers the advantage of circumventing direct data sharing, as clients may exchange generative models or synthetic representations instead of raw data, which strengthens privacy preservation (Salami et al., 2024; Kang et al., 2025; Elmas et al., 2022; Aouedi et al., 2025). For example, LANDER exploits pretrained label-text embeddings as semantic anchors for client training and generation (Tran et al., 2024); HR leverages an autoencoder to replay latent representations and globally align class centroids, while decoding perturbed embeddings to generate unseen-task samples (Nori et al., 2025). However, these methods still face several challenges, including the catastrophic for-

getting of generators themselves, limited quality and diversity of synthesized samples, substantial computational and communication overhead in federated settings.

## 2.2 EXEMPLAR REPLAY-BASED METHODS

To address the aforementioned limitations of generative replay, exemplar replay-based methods have been proposed, which maintain a small memory buffer of real samples from previously learned classes, and rehearse them jointly with new task data to reinforce prior knowledge and alleviate catastrophic forgetting (Dong et al., 2022; Li et al., 2024b; Lu et al., 2024). By explicitly preserving authentic distributions, these methods mitigate the challenges of limited generative fidelity and semantic inconsistency, while obviating the need for costly generator training (Li et al., 2025). For instance, Re-Fed+ selects and caches critical old samples based on importance measures derived from per-sample training gradients of the personalized informative model at each client (Li et al., 2025); FedCBDR reconstructs global pseudo features and performs class-balanced sampling based on their leverage scores (Qi et al., 2025). However, these methods typically lack rigorous theoretical guarantees for importance-driven selection, and remain limited in their ability to leverage multi-view information under strict privacy constraints. This study aims to alleviate these issues.

## 3 METHODOLOGY

### 3.1 PRELIMINARIES

We consider a FCIL system with $K$ clients. For each new task $t$, client $k$ constructs a replay buffer by sampling up to $N$ exemplars from each previously task $\{1, \ldots, t-1\}$:

$$\mathcal{B}_k^{(t-1)} = \bigcup_{s=1}^{t-1} \{(x_{k,i}^{(s)}, y_{k,i}^{(s)})\}_{i=1}^N, \quad \text{with } |\mathcal{B}_k^{(t-1)}| \leq M. \tag{1}$$

This buffer is integrated with the current task data $\mathcal{D}_k^{(t)}$ to form the augmented local training set $\mathcal{D}_{k,\text{train}}^{(t)} = \mathcal{D}_k^{(t)} \cup \mathcal{B}_k^{(t-1)}$. Local models are subsequently optimized on this enriched dataset, enabling them to assimilate novel knowledge while mitigating the erosion of prior information. During global aggregation, the server updates the shared parameters by minimizing the cumulative empirical risk: $\min_\theta \sum_{k=1}^K \sum_{(x,y) \in \mathcal{D}_{k,\text{train}}^{(t)}} \mathcal{L}(h_k(x;\theta), y)$, where $h_k(\cdot;\theta)$ denotes the local model of client $k$, and $\mathcal{L}(\cdot)$ represents the classification objective. In the following, when the context is clear, we omit the superscript of task $t$ of the data for simplicity. In addition, we consider each local model a neural network composed of a feature extractor (backbone) and a linear prediction head followed by a fixed activation function $f_k(x;\theta_k) = f(\phi_k(x)^\top \theta_k)$, where $\phi_k : \mathcal{X} \to \mathbb{R}^d$ is the backbone and $\theta_k \in \mathbb{R}^d$ is the linear head. We assume throughout that the activation $f : \mathbb{R} \to \mathbb{R}$ is $L$-Lipschitz and is applied coordinatewise with $f(0) = 0$ (for example, ReLU, tanh). For client $k$, stack backbone features of $n_k$ local samples into the matrix

$$A_k = [a_{k,1}^\top, \ldots, a_{k,n_k}^\top]^\top \in \mathbb{R}^{n_k \times d}, \qquad a_{k,i} = \phi_k(x_{k,i}) \in \mathbb{R}^d, \tag{2}$$

so the model's prediction vector on client $k$ is $f(A_k \theta_k)$. We assume $d \ll n_k$ and $A_k$ has full column rank. For multi-dimensional target, we may use one-vs-rest heads; without loss of generality, the analysis below focuses on a single response vector for simplicity.

### 3.2 MOTIVATION AND HIGH-LEVEL IDEA

Exemplar replay-based FCIL aims to select a compact, representative subset that retains global knowledge while mitigating forgetting. Existing approaches mainly suffer from two limitations: (1) Local-view selection mines exemplars via local heuristics but lack global awareness, limiting their global contribution; and (2) Global-view selection accounts for cross-client distributions via aggregated feature statistics, but may incur privacy risks by exposing feature statistics and spectral information, as well as substantial communication overhead.

To address these problems, we propose to utilize Lewis weight sampling (Fazel et al., 2022), which provably preserves the operator norm of the feature map. By approximately preserving the Gram

matrix, i.e., $X_{\text{subset}}^\top X_{\text{subset}} \approx X_{\text{full}}^\top X_{\text{full}}$ in a spectral sense, the selected exemplars retain the principal directions of variance of the full data, which helps prevent old decision boundaries from collapsing or drifting arbitrarily when new classes are introduced and ensures that worst-case, tail-region directions remain represented in the replay buffer in the FCIL setting. We briefly recall the $\ell_p$ Lewis weights and the subspace-embedding guarantee induced by sampling rows proportionally to an overestimate of these weights.

**Definition 1** (Lewis Weight Fazel et al. (2022)). *Let $A \in \mathbb{R}^{n \times d}$ have rows $a_i^\top$ and let $p \geq 1$. The $\ell_p$ Lewis weights $w_i = w_i(A)$ are defined implicitly by*

$$w_i = \left( a_i^\top \left( A^\top W^{1-\frac{2}{p}} A \right)^{-1} a_i \right)^{p/2}, \quad W = \text{diag}(w_1, \ldots, w_n). \tag{3}$$

*They satisfy $w_i \in [0,1]$ and $\sum_{i=1}^n w_i = d$. For $p = 2$, Lewis weights coincide with leverage scores $w_i = a_i^\top (A^\top A)^{-1} a_i$.*

**Definition 2** ($\ell_p$ subspace embedding). *A matrix $S \in \mathbb{R}^{m \times n}$ is an $\ell_p$ $\varepsilon$-subspace embedding for $A \in \mathbb{R}^{n \times d}$ if, simultaneously for all $\theta \in \mathbb{R}^d$,*

$$(1 - \varepsilon)\|A\theta\|_p \leq \|SA\theta\|_p \leq (1 + \varepsilon)\|A\theta\|_p. \tag{4}$$

**Definition 3** (Reweighted sampling matrix). *Given probabilities $(p_1, \ldots, p_n)$ summing to 1, construct $S \in \mathbb{R}^{m \times n}$ by drawing each row i.i.d. as $X = (mp_j)^{-1/p} e_j^\top$ with probability $p_j$. We call $S$ a reweighted sampling matrix (with sample size $m$).*

**Lemma 1** (Constant-factor subspace embedding (Cohen & Peng, 2015)). *Let $A \in \mathbb{R}^{n \times d}$, $p \geq 1$, and suppose $t_i \geq \beta \, w_i(A)$ with oversampling parameter*

$$\beta \gtrsim_p \begin{cases} \log^3 d + \log \frac{1}{\delta} & 0 < p < 2, \, p \neq 1, \\ \log \frac{d}{\delta} & p = 1, 2, \\ d^{\frac{p}{2}-1}\left(\log d + \log \frac{1}{\delta}\right) & 2 < p < \infty. \end{cases}$$

*Let $m = \sum_i t_i$ and let $S$ be the reweighted sampling matrix with sampling probability $p_i = t_i/m$. Then, with probability at least $1 - \delta$, $S$ is an $\ell_p$ $(1/2)$-subspace embedding for $A$.*

In exemplar replay, we seek a small subset whose induced objective is a faithful surrogate for training on the full data. Geometrically, faithfulness means that the subset preserves how the feature matrix $A$ acts on all directions $\theta \in \mathbb{R}^d$. This is precisely what $\ell_p$ Lewis-weight sampling guarantees: the sampled-and-reweighted matrix $SA$ is a subspace embedding for $A$, implying that the "energy" $\|A\theta\|_p$ is preserved up to a $(1 \pm \varepsilon)$ factor for every $\theta$. In the $p = 2$ case, this is equivalent to preserving the Gram operator $A^\top A$ in the sense that

$$(1-\varepsilon)\|A\theta\|_2^2 \leq \|SA\theta\|_2^2 \leq (1+\varepsilon)\|A\theta\|_2^2 \iff \|A^\top A - (SA)^\top (SA)\|_2 \leq c\varepsilon\|A\|_2^2, \tag{5}$$

where $c$ is a universal constant. Preserving $A^\top A$ in operator norm stabilizes the condition number seen by the head, so gradient directions and step sizes computed on exemplars remain faithful to those on full data. This reduces optimization bias due to replay.

Furthermore, with $f$ being $L$-Lipschitz and $f(0) = 0$, for any $\theta$, it holds that

$$\|f(A\theta) - f(SA\theta)\|_p \leq L\|A\theta - SA\theta\|_p, \tag{6}$$

so subspace embeddings transfer directly into uniform control of prediction errors across all heads $\theta$. By further employing a recent theoretical study that considers the activation function in Lewis weight sampling (Huang et al., 2024), we can bound the errors of the model trained on the sampled data. Please see the theoretical analysis in Sec. 3.5.

### 3.3 CROSS-VIEW LEWIS WEIGHT FUSION FOR FEDERATED EXEMPLAR SAMPLING

In federated learning, single-view estimation may be biased due to heterogeneity. The cross-view Lewis weights (CV-LWF) module seeks a replay sampling distribution that is simultaneously safe for multiple representations by aggregating multiple model perspectives. To achieve this goal, we are inspired by (Huang et al., 2024) that sampling with probabilities proportional to any overestimate of the true $\ell_p$ Lewis weights preserves the subspace embedding, according to Lemma 1. Therefore,

Figure 2: Illustration of the `CLIF` framework, exemplified by the first two tasks. It first employs cross-view Lewis weight fusion after completing the training of Task 1 to comprehensively assess sample importance. Subsequently, it adjusts sample weights according to their sampling frequency to emphasize the contribution of important samples in subsequent tasks.

if a fused score upper-bounds the per-view Lewis weight for every view, the resulting sample-and-reweight procedure guarantees subspace preservation, and hence the error bounds, for each model view at once.

Formally, each client $k$ forms a small set of views (representations) by passing its local samples through multiple available backbones:

$$\mathcal{V}_k = \big\{\phi^{(m)}\big\}_{m=1}^M, \quad A_k^{(m)} \in \mathbb{R}^{n_k \times d} \text{ with rows } a_{k,i}^{(m)} = \phi^{(m)}(x_{k,i}). \tag{7}$$

These backbones may come from other clients. Note that transferring model between clients is usually acceptable in federated learning (Zhang et al., 2021; Luo & Wu, 2022), the communication and bandwidth overhead is provided in Sec. E.7. And we also provide two alternative approaches to mitigate privacy leakage in Sec. E.8, namely adding Gaussian perturbation and employing a dynamic Beta-weighted combination across different client models.

For each view $m \in [M]$, compute the $\ell_p$ Lewis weights $w_{k,i}^{(p,m)} = w_i\big(A_k^{(m)}\big)$, and define the fused score by the pointwise maximum $\tilde{w}_{k,i} = \max_{m \in [M]} w_{k,i}^{(p,m)}$. Set sampling weights $t_{k,i} \propto \tilde{w}_{k,i}$ and probabilities $p_{k,i} = t_{k,i}/\sum_j t_{k,j}$. By construction, $t_{k,i} \geq \beta\, w_{k,i}^{(p,m)}$ for every $m$, so the sampled/reweighted subset simultaneously preserves all represented subspaces. More concretely, consider the objective on client $k$ and view $m$ with label vector $y_k$:

$$\theta_{k,*}^{(m)} = \arg\min_{\theta \in \mathbb{R}^d} \|f(A_k^{(m)}\theta) - y_k\|_p^p. \tag{8}$$

CV-LWF draws $m_k$ samples i.i.d. with replacement from $\{1,\dots,n_k\}$ using $(p_{k,i})$ for replay. Repeatedly sampled instances only affect their training weights and do not consume additional budget.

### 3.4 FREQUENCY-AWARE WEIGHTED TRAINING (FWT)

This section introduces the reweighting mechanism on the sampled data. Let $Q_k$ be the set of distinct indices drawn for the replay buffer $\mathcal{B}_k$; set $|Q_k| \leq N$. Following the theory, build the reweighted sampling matrix $S_k$ as in Definition 3, i.e., each sampled row $i \in Q_k$ is assigned weight $(m_k\, p_{k,i})^{-1/p}$ in $S_k$ so that $S_k A_k^{(m)}$ becomes an $\ell_p$ subspace surrogate of $A_k^{(m)}$ for all $m$. Consequently, the error of the solution of the following reweighted problem is bounded:

$$\tilde{\theta}_k^{(m)} = \arg\min_{\theta \in E_k^{(m)}} \left\| S_k f\big(A_k^{(m)}\theta\big) - S_k y_k \right\|_p^p, \qquad E_k^{(m)} \triangleq \left\{\theta : \|S_k A_k^{(m)}\theta\|_p^p \leq \frac{\|S_k y_k\|_p^p}{\varepsilon\, L^p}\right\}, \tag{9}$$

where $E_k^{(m)}$ is the standard radius control for analysis, and is typically omitted in practice. And we present the performance impact analysis of different re-weighting loss functions in Sec. E.10.

### 3.5 THEORETICAL ANALYSIS

Our cross-view sampling method CV-LWF attains the following bound on every view simultaneously. The proof is deferred to appendix.

---

**Algorithm 1** `CLIF`

---

1: **Initialize:** $R$: communication rounds; $K$: number of clients; $T$: number of tasks; $\mathcal{B}_k^{\text{pre}}$: client $k$'s replay buffer of past tasks; $\mathcal{D}_k^s$: client $k$'s local dataset for task $s$; $\theta_g$: global model parameters; $\theta_k$: client $k$'s local model parameters.
2: **for** task $t = 1$ to $T$ **do**
3:   **for** communication round $r = 1$ to $R$ **do**
4:     **for** each client $k = 1$ to $K$ **do**
5:       Initialize local model parameters: $\theta_k \leftarrow \theta_g$
6:       **if** $t == 1$ **then**
7:         Draw $\zeta$ from $\mathcal{D}_k^{(1)}$ and apply Cross Entropy loss to update $\theta_k$.
8:       **else**
9:         Draw $\zeta$ from $\mathcal{D}_k^{(s)} \cup \mathcal{B}_k^{pre}$, and update $\theta_k$ using Eq. 9.
10:       **end if**
11:     **end for**
12:     **if** $r < R$ **then**
13:       Compute the global model by aggregating $\theta_k$ across clients.
14:     **else**
15:       Randomly sample $k$ other-client models for each client and send them to that client.
16:       Compute cross-view Lewis weights under the different models (Eq. 3), and fuse them across views using the max rule (see Line 245).
17:       Execute sampling based on fused Lewis weights on client $k$ to obtain exemplars $\mathcal{E}_k$ and store them in $\mathcal{B}_k^{\text{pre}}$.
18:       Compute the global model by aggregating $\theta_k$ across clients.
19:     **end if**
20:   **end for**
21: **end for**

---

**Theorem 1.** *Fix a client $k$ and task $t$. For each view $m \in [M]$, let $w_i(A_k^{(m)})$ be the $\ell_p$ Lewis weight of row $i$ and define the fused scores $\tilde{w}_{k,i} = \max_{m \in [M]} w_i(A_k^{(m)})$. Let $T_k = \sum_{i=1}^{n_k} \tilde{w}_{k,i}$. Build a reweighted sampling matrix $S_k$ by drawing $m_k$ rows i.i.d. with probabilities $p_{k,i} = \tilde{w}_{k,i}/T_k$ and per-row weight $(m_k p_{k,i})^{-1/p}$. For any activation functions $f$ that is $L$-Lipschitz with $f(0) = 0$ and $p \geq 1$. Consider the constrained, reweighted objective on $\mathcal{D}_{k,train}^{(t)}$ for each view $m$:*

$$\tilde{\theta}_k^{(m)} \in \arg\min_{\theta \in E_k^{(m)}} \left\| S_k f(A_k^{(m)} \theta) - S_k y_k \right\|_p^p, \qquad E_k^{(m)} = \left\{ \theta : \left\| S_k A_k^{(m)} \theta \right\|_p^p \leq \frac{\|S_k y_k\|_p^p}{\varepsilon L^p} \right\}. \quad (10)$$

*If $m_k \gtrsim_p \varepsilon^{-4} T_k \, d^{\max\{\frac{p}{2}-1,0\}} \log^2 d \cdot \log\left(\frac{dT_k}{\varepsilon}\right)$, then, $\forall m \in [M]$, the following holds with probability at least $0.9$,*

$$\left\| f(A_k^{(m)} \tilde{\theta}_k^{(m)}) - y_k \right\|_p^p \leq C_p \left( \left\| f(A_k^{(m)} \theta_{k,*}^{(m)}) - y_k \right\|_p^p + \varepsilon L^p \left\| A_k^{(m)} \theta_{k,*}^{(m)} \right\|_p^p \right), \quad (11)$$

*where $\theta_{k,*}^{(m)} \in \arg\min_\theta \| f(A_k^{(m)} \theta) - y_k \|_p^p$ and $C_p > 0$ is a constant depending only on $p$.*

**Remark.** The bound on $m_k$ scales with $T_k$, which is upper-bounded by $Md$ in the worst case. Replay budgets in our experiments are much smaller than this bound; thus the guarantee is primarily qualitative. Since each client's local heads $\{\tilde{\theta}_k^{(m)}\}_m$ trained on the reweighted exemplars satisfy the per-view error bound, their empirical risks are uniformly controlled. Under standard FL aggregation (e.g., FedAvg), weighted averages of locally well-approximated heads yield a global model whose risk reflects these local bounds; hence we expect improved stability across tasks and reduced forgetting when aggregating from exemplar-replay clients. We emphasize that Theorem 1 is proved for $\ell_p$ objectives. Nevertheless, the same sampling rule applies naturally to multi-class classification trained with cross-entropy, a point that has been empirically validated in prior work (Huang et al., 2024). Intuitively, cross-entropy acts as a smooth, classification-calibrated surrogate whose gradients depend on the underlying feature geometry; thus, preserving subspace structure via Lewis-weight sampling remains beneficial. Accordingly, in our experiments we adopt the proposed sampling strategy for general federated learning tasks beyond the $\ell_p$ setting.

Table 1: Performance comparison between CLIF and baselines across three datasets under varying levels of heterogeneity ($\beta = \{0.5, 1.0\}$). Two task-split settings are adopted: 3/5/5 and 5/10/10, where CIFAR10, CIFAR100, and Tiny-ImageNet are split into 3/5, 5/10, and 5/10 tasks, respectively. Results are averaged over three random seeds, and we report mean±standard deviation. The best results are highlighted in **bold**.

| | CIFAR10 | | CIFAR100 | | Tiny-ImageNet | | CIFAR10 | | CIFAR100 | | Tiny-ImageNet | |
|---|---|---|---|---|---|---|---|---|---|---|---|---|
| | **20 Clients (3/5/5 Task-Splits)** | | | | | | **40 Clients (3/5/5 Task-Splits)** | | | | | |
| | $\beta=0.5$ | $\beta=1.0$ | $\beta=0.5$ | $\beta=1.0$ | $\beta=0.5$ | $\beta=1.0$ | $\beta=0.5$ | $\beta=1.0$ | $\beta=0.5$ | $\beta=1.0$ | $\beta=0.5$ | $\beta=1.0$ |
| FineTune | $27.89_{\pm2.7}$ | $37.71_{\pm2.1}$ | $16.07_{\pm1.2}$ | $17.81_{\pm0.8}$ | $9.59_{\pm2.3}$ | $10.11_{\pm1.5}$ | $31.02_{\pm1.9}$ | $38.74_{\pm2.4}$ | $15.48_{\pm1.8}$ | $17.84_{\pm2.0}$ | $10.46_{\pm2.1}$ | $10.71_{\pm1.6}$ |
| FedEWC | $28.33_{\pm1.4}$ | $37.82_{\pm1.9}$ | $17.45_{\pm2.4}$ | $19.08_{\pm2.1}$ | $10.34_{\pm1.7}$ | $11.49_{\pm2.5}$ | $32.31_{\pm1.3}$ | $39.46_{\pm1.6}$ | $16.79_{\pm1.9}$ | $19.43_{\pm1.8}$ | $11.16_{\pm2.0}$ | $11.25_{\pm1.4}$ |
| FedLwf | $33.63_{\pm1.2}$ | $42.93_{\pm1.8}$ | $24.02_{\pm1.7}$ | $30.71_{\pm1.9}$ | $14.85_{\pm2.1}$ | $16.45_{\pm1.6}$ | $42.05_{\pm2.3}$ | $44.20_{\pm1.1}$ | $26.64_{\pm1.5}$ | $31.96_{\pm2.4}$ | $15.04_{\pm0.9}$ | $15.30_{\pm1.3}$ |
| Target | $29.95_{\pm1.8}$ | $38.82_{\pm2.2}$ | $19.85_{\pm1.0}$ | $23.67_{\pm2.5}$ | $13.74_{\pm1.9}$ | $15.38_{\pm1.2}$ | $32.76_{\pm0.7}$ | $34.77_{\pm1.5}$ | $22.45_{\pm2.0}$ | $24.81_{\pm1.8}$ | $14.79_{\pm2.3}$ | $15.11_{\pm1.6}$ |
| LANDER | $32.66_{\pm1.9}$ | $39.71_{\pm2.0}$ | $22.73_{\pm1.1}$ | $33.69_{\pm2.3}$ | $16.15_{\pm0.6}$ | $17.46_{\pm1.4}$ | $36.15_{\pm1.7}$ | $39.89_{\pm2.5}$ | $38.76_{\pm2.1}$ | $43.43_{\pm1.2}$ | $15.89_{\pm2.4}$ | $16.39_{\pm1.8}$ |
| FCIDF | $51.12_{\pm2.3}$ | $52.37_{\pm1.0}$ | $33.58_{\pm2.1}$ | $39.85_{\pm1.9}$ | $15.29_{\pm1.9}$ | $18.45_{\pm1.6}$ | $55.18_{\pm2.1}$ | $60.88_{\pm1.2}$ | $39.28_{\pm2.5}$ | $40.17_{\pm0.8}$ | $14.01_{\pm1.3}$ | $15.56_{\pm2.4}$ |
| Re-Fed+ | $52.45_{\pm1.9}$ | $61.74_{\pm1.1}$ | $42.79_{\pm2.3}$ | $43.56_{\pm1.4}$ | $18.88_{\pm2.0}$ | $19.46_{\pm1.6}$ | $67.16_{\pm2.5}$ | $69.42_{\pm1.3}$ | $45.12_{\pm1.8}$ | $46.04_{\pm0.9}$ | $18.28_{\pm2.7}$ | $18.89_{\pm1.8}$ |
| CLIF$_R$ | $57.30_{\pm1.2}$ | $64.29_{\pm1.3}$ | $\mathbf{44.10}_{\pm1.1}$ | $45.83_{\pm1.6}$ | $\mathbf{19.47}_{\pm0.6}$ | $\mathbf{19.96}_{\pm1.6}$ | $69.44_{\pm1.2}$ | $72.10_{\pm2.1}$ | $46.65_{\pm1.2}$ | $46.40_{\pm1.3}$ | $\mathbf{19.86}_{\pm1.1}$ | $\mathbf{20.27}_{\pm1.5}$ |
| FedCBDR | $56.94_{\pm1.6}$ | $66.01_{\pm2.2}$ | $42.52_{\pm1.5}$ | $44.85_{\pm0.8}$ | $18.81_{\pm1.6}$ | $19.79_{\pm1.2}$ | $70.65_{\pm2.3}$ | $72.57_{\pm1.6}$ | $45.74_{\pm2.4}$ | $46.20_{\pm1.1}$ | $18.92_{\pm2.3}$ | $19.24_{\pm2.1}$ |
| CLIF$_F$ | $\mathbf{62.47}_{\pm2.4}$ | $\mathbf{68.29}_{\pm1.0}$ | $43.84_{\pm1.7}$ | $\mathbf{46.34}_{\pm1.2}$ | $19.14_{\pm0.9}$ | $19.60_{\pm1.8}$ | $\mathbf{73.23}_{\pm1.5}$ | $\mathbf{76.21}_{\pm2.0}$ | $\mathbf{46.86}_{\pm2.1}$ | $\mathbf{47.26}_{\pm2.0}$ | $19.79_{\pm1.4}$ | $19.99_{\pm1.3}$ |
| | **20 Clients (5/10/10 Task-Splits)** | | | | | | **40 Clients (5/10/10 Task-Splits)** | | | | | |
| | $\beta=0.5$ | $\beta=1.0$ | $\beta=0.5$ | $\beta=1.0$ | $\beta=0.5$ | $\beta=1.0$ | $\beta=0.5$ | $\beta=1.0$ | $\beta=0.5$ | $\beta=1.0$ | $\beta=0.5$ | $\beta=1.0$ |
| FineTune | $20.45_{\pm1.4}$ | $23.63_{\pm0.9}$ | $10.42_{\pm1.7}$ | $11.67_{\pm1.2}$ | $6.05_{\pm0.7}$ | $6.47_{\pm1.9}$ | $23.55_{\pm1.6}$ | $24.84_{\pm1.1}$ | $12.21_{\pm1.9}$ | $17.84_{\pm0.8}$ | $6.22_{\pm1.3}$ | $6.94_{\pm1.5}$ |
| FedEWC | $21.36_{\pm1.2}$ | $16.97_{\pm0.6}$ | $10.50_{\pm1.8}$ | $12.61_{\pm1.1}$ | $6.89_{\pm1.4}$ | $7.45_{\pm1.9}$ | $24.12_{\pm1.0}$ | $25.29_{\pm1.7}$ | $13.94_{\pm0.9}$ | $19.43_{\pm1.6}$ | $6.61_{\pm1.3}$ | $7.58_{\pm1.5}$ |
| FedLwf | $27.46_{\pm1.1}$ | $29.94_{\pm1.8}$ | $15.71_{\pm1.3}$ | $20.97_{\pm0.7}$ | $12.80_{\pm1.6}$ | $13.44_{\pm1.4}$ | $30.06_{\pm1.2}$ | $31.96_{\pm1.5}$ | $15.41_{\pm1.7}$ | $21.19_{\pm0.9}$ | $13.79_{\pm1.8}$ | $15.05_{\pm0.6}$ |
| Target | $22.21_{\pm1.3}$ | $25.38_{\pm1.4}$ | $13.89_{\pm0.8}$ | $18.02_{\pm1.2}$ | $8.47_{\pm1.9}$ | $9.26_{\pm1.0}$ | $26.31_{\pm1.5}$ | $30.37_{\pm1.1}$ | $13.24_{\pm1.8}$ | $15.24_{\pm1.6}$ | $10.47_{\pm0.7}$ | $11.28_{\pm1.4}$ |
| LANDER | $23.64_{\pm0.9}$ | $32.78_{\pm1.7}$ | $16.20_{\pm1.3}$ | $21.46_{\pm1.1}$ | $9.37_{\pm1.6}$ | $16.09_{\pm1.9}$ | $27.17_{\pm1.0}$ | $43.40_{\pm1.8}$ | $17.46_{\pm1.5}$ | $21.38_{\pm0.6}$ | $14.80_{\pm1.2}$ | $16.21_{\pm1.7}$ |
| FCIDF | $46.84_{\pm1.5}$ | $47.16_{\pm1.9}$ | $33.37_{\pm1.2}$ | $35.20_{\pm1.3}$ | $14.34_{\pm0.8}$ | $16.43_{\pm1.4}$ | $52.33_{\pm1.6}$ | $56.09_{\pm0.9}$ | $42.84_{\pm1.7}$ | $44.26_{\pm1.2}$ | $13.30_{\pm1.3}$ | $14.74_{\pm2.2}$ |
| Re-Fed+ | $49.10_{\pm1.7}$ | $60.32_{\pm1.4}$ | $39.18_{\pm0.9}$ | $40.42_{\pm1.3}$ | $15.19_{\pm1.8}$ | $16.45_{\pm1.1}$ | $61.92_{\pm1.2}$ | $68.28_{\pm1.6}$ | $52.15_{\pm1.4}$ | $53.20_{\pm1.7}$ | $16.26_{\pm0.6}$ | $16.68_{\pm1.5}$ |
| CLIF$_R$ | $52.34_{\pm0.8}$ | $63.04_{\pm1.2}$ | $40.23_{\pm1.5}$ | $41.05_{\pm1.9}$ | $16.27_{\pm1.0}$ | $17.51_{\pm1.6}$ | $63.44_{\pm1.7}$ | $71.34_{\pm1.1}$ | $53.26_{\pm1.8}$ | $53.88_{\pm0.7}$ | $\mathbf{18.04}_{\pm2.1}$ | $\mathbf{18.32}_{\pm1.3}$ |
| FedCBDR | $50.50_{\pm1.6}$ | $65.47_{\pm0.9}$ | $39.37_{\pm1.3}$ | $39.52_{\pm1.7}$ | $15.38_{\pm1.2}$ | $17.40_{\pm1.8}$ | $66.78_{\pm1.4}$ | $74.05_{\pm1.0}$ | $51.09_{\pm1.9}$ | $52.30_{\pm1.1}$ | $16.08_{\pm0.7}$ | $16.11_{\pm1.5}$ |
| CLIF$_F$ | $\mathbf{52.85}_{\pm1.2}$ | $\mathbf{68.14}_{\pm1.9}$ | $\mathbf{40.42}_{\pm1.1}$ | $\mathbf{41.39}_{\pm1.4}$ | $\mathbf{16.49}_{\pm1.8}$ | $\mathbf{17.63}_{\pm1.5}$ | $\mathbf{68.06}_{\pm0.9}$ | $\mathbf{75.54}_{\pm1.3}$ | $\mathbf{53.30}_{\pm1.6}$ | $\mathbf{54.52}_{\pm1.0}$ | $17.14_{\pm1.3}$ | $17.18_{\pm1.7}$ |

## 4 EXPERIMENTS

### 4.1 EXPERIMENT SETTINGS

**Datasets.** Following existing methods (Li et al., 2025; Qi et al., 2025), we evaluate the effectiveness of the proposed method on three commonly used benchmark datasets, including CIFAR10 (Krizhevsky et al., 2009), CIFAR100 (Krizhevsky et al., 2009), and Tiny-ImageNet (Le & Yang, 2015).

**Evaluation Metric.** Following existing studies (Li et al., 2025; Qi et al., 2025), we use Top-1 accuracy (Top-1 Acc) to evaluate the performance of all methods.

**Implementation Details.** For fair comparison, all methods adopt ResNet-18 as the backbone, with the classification head expanded for each new task. Experiments are conducted with 20 or 40 clients, 20% of which are sampled per round for 200 communication rounds. Data heterogeneity is simulated with Dirichlet distributions ($\beta = \{0.5, 1.0\}$). CIFAR10 is split into 3 and 5 tasks, while CIFAR100 and Tiny-ImageNet are partitioned into 5 and 10 tasks, respectively. Each client is trained with local epochs $E = 2$. The batch size is set to $B = 128$ for 20 clients and $B = 64$ for 40 clients. We employ the SGD optimizer with a learning rate of 0.01 and a weight decay of $1 \times 10^{-5}$. $p = \{1, 2\}$ and $m = \{2, 5, 10\}$ for multi-view Lewis weight calculation. Each client maintains a fixed-size exemplar memory for replay. When using 20 clients, each client stores 60 samples per task for CIFAR10 (3 tasks), 40 for CIFAR10 (5 tasks), 40 for CIFAR100 (5 tasks), 20 for CIFAR100 (10 tasks), 80 for Tiny-ImageNet (5 tasks), and 40 for Tiny-ImageNet (10 tasks). For experiments with 40 clients, the number of exemplars per task is halved accordingly. The hyperparameter settings of baselines follow their original papers to ensure fairness: (i) baselines with established federated versions using their original configurations, and (ii) baselines without original federated versions using widely adopted public implementations.

Table 2: Ablation results of CV-LWF and FWT on CIFAR10/100 with 5-task splits, 20 clients (20% participation), using Re-Fed+ and FedCBDR as bases under $\beta = 0.5$ and $\beta = 1.0$.

| | Base (Re-Fed+) | | | | Base (FedCBDR) | | | |
|---|---|---|---|---|---|---|---|---|
| | CIFAR10 | | CIFAR100 | | CIFAR10 | | CIFAR100 | |
| | $\beta = 0.5$ | $\beta = 1.0$ | $\beta = 0.5$ | $\beta = 1.0$ | $\beta = 0.5$ | $\beta = 1.0$ | $\beta = 0.5$ | $\beta = 1.0$ |
| Base | $52.45_{\pm 1.9}$ | $61.74_{\pm 1.1}$ | $42.79_{\pm 2.3}$ | $43.56_{\pm 1.4}$ | $56.94_{\pm 1.6}$ | $66.01_{\pm 2.2}$ | $42.52_{\pm 1.5}$ | $44.85_{\pm 0.8}$ |
| + CV-LWF ($p = 1$) | $56.11_{\pm 2.0}$ | $63.37_{\pm 2.2}$ | $43.61_{\pm 0.6}$ | $45.19_{\pm 0.8}$ | $60.94_{\pm 0.9}$ | $67.46_{\pm 2.2}$ | $43.21_{\pm 2.9}$ | $46.02_{\pm 2.9}$ |
| + CV-LWF ($p = 2$) | $56.40_{\pm 1.4}$ | $63.52_{\pm 2.5}$ | $43.82_{\pm 1.6}$ | $45.27_{\pm 1.2}$ | $61.68_{\pm 2.2}$ | $67.11_{\pm 1.1}$ | $43.47_{\pm 2.5}$ | $45.68_{\pm 2.1}$ |
| + CV-LWF ($p = 1$) + FWT | $\mathbf{57.30}_{\pm 1.2}$ | $64.13_{\pm 1.8}$ | $\mathbf{44.10}_{\pm 1.1}$ | $45.76_{\pm 2.7}$ | $62.28_{\pm 2.7}$ | $\mathbf{68.29}_{\pm 1.0}$ | $43.54_{\pm 0.8}$ | $\mathbf{46.34}_{\pm 1.2}$ |
| + CV-LWF ($p = 2$) + FWT | $57.11_{\pm 1.7}$ | $\mathbf{64.29}_{\pm 1.3}$ | $44.07_{\pm 1.2}$ | $\mathbf{45.83}_{\pm 1.6}$ | $\mathbf{62.47}_{\pm 2.4}$ | $67.47_{\pm 1.0}$ | $\mathbf{43.84}_{\pm 1.7}$ | $46.12_{\pm 1.8}$ |

## 4.2 PERFORMANCE COMPARISON

This section compares the overall performance of the proposed `CLIF` against eight state-of-the-art approaches, including FineTune (Tran et al., 2024), FedEWC (Kirkpatrick et al., 2017), FedLwf (Li & Hoiem, 2017), Target (Zhang et al., 2023), LANDER (Tran et al., 2024), FCIDF (Lu et al., 2024), Re-Fed+ (Li et al., 2025) and FedCBDR (Qi et al., 2025). We introduce two variants of our framework: $\text{CLIF}_R$ computes multi-view Lewis scores locally on each client and selects replay samples on the client side, following the local replay mechanism of Re-Fed+. $\text{CLIF}_F$ aggregates pseudo samples from all clients on the server, computes multi-view Lewis scores centrally to obtain a global ranking of replay candidates, following the centralized replay pipeline of FedCBDR. The detailed procedure is outlined in Algorithm 1. Table 1 reports the following comparative results:

- Both variants of `CLIF` ($\text{CLIF}_R$ and $\text{CLIF}_F$) consistently rank among the top two performers in the most of cases, while each variant yields clear improvements over all baseline methods.

- Across different degrees of heterogeneity, `CLIF` consistently maintains its advantage even as the number of clients increases from 20 to 40, demonstrating robustness to skewed client distributions and scalability to larger federated networks.

- Knowledge-distillation-based methods (FedLwf, Target) achieve moderate improvements compared to FineTune, but their gains are unstable. Especially under high heterogeneity, their performance collapses, showing that distillation is insufficient for long-term knowledge retention.

- LANDER generally performs well across many cases, but its performance is unstable and consistently inferior to `CLIF`, particularly on CIFAR100. This instability stems from the forgetting issue of its generator, which weakens the quality of replayed data.

## 4.3 ABLATION STUDY

This section evaluates the contributions of the main modules, including CV-LWF, FWT. The key findings are reported in Table 2 and can be summarized as follows:

- Integrating the Cross-View Lewis Weight Fusion (CV-LWF) module consistently improves the performance of both base methods across all settings, indicating its effectiveness in enhancing exemplar representativeness and stabilizing replay selection.

- Incorporating the Frequency-aware Weighted Training (FWT) module further enhances performance, suggesting that the manner in which replay data are utilized plays a crucial role in mitigating forgetting and improving generalization.

- The integration of both modules achieves the best performance, as it simultaneously enhances exemplar representativeness and optimizes their utilization during training.

## 4.4 QUANTITATIVE ANALYSIS UNDER DIFFERENT HYPERPARAMETER SETTINGS

This section explores the performance of $\text{CLIF}_R$, $\text{CLIF}_F$, FCIDF, Re-Fed+, and FedCBDR under diverse hyperparameter settings, covering different memory buffer sizes, local training epochs, fraction of clients participating per round, and varying numbers of views for data selection.

Table 3: Evaluation of the two CLIF variants compared with Re-Fed+ and FedCBDR under different $m$-view settings ($m = \{2, 5, 10\}$) across three datasets.

| | CIFAR10 | | | CIFAR100 | | | Tiny-ImageNet | | |
|---|---|---|---|---|---|---|---|---|---|
| | $m=2$ | $m=5$ | $m=10$ | $m=2$ | $m=5$ | $m=10$ | $m=2$ | $m=5$ | $m=10$ |
| FCIDF | | $46.84_{\pm1.5}$ | | | $33.58_{\pm2.1}$ | | | $15.29_{\pm1.9}$ | |
| Re-Fed+ | | $49.10_{\pm1.7}$ | | | $42.79_{\pm2.3}$ | | | $18.28_{\pm2.7}$ | |
| CLIF$_R$ | $52.34_{\pm0.8}$ | $54.97_{\pm1.4}$ | $51.66_{\pm1.9}$ | $44.10_{\pm1.1}$ | $45.35_{\pm2.1}$ | $44.55_{\pm1.7}$ | $19.45_{\pm1.2}$ | $19.86_{\pm1.4}$ | $19.67_{\pm1.1}$ |
| FedCBDR | | $50.50_{\pm1.6}$ | | | $42.52_{\pm1.5}$ | | | $18.92_{\pm2.3}$ | |
| CLIF$_F$ | $52.85_{\pm1.2}$ | $56.63_{\pm1.9}$ | $57.67_{\pm1.8}$ | $43.84_{\pm1.4}$ | $44.20_{\pm1.7}$ | $44.28_{\pm1.6}$ | $19.63_{\pm1.3}$ | $19.79_{\pm1.3}$ | $19.19_{\pm1.5}$ |

**Data Selection under Varying Numbers of Views.** We evaluate the effect of varying $m$-views on performance across three datasets, with $m \in \{2, 5, 10\}$. As reported in Table 3, as the $m$ value increases from 2 to 5, the model performance improves accordingly. This is reasonable, as multi-view features extracted from more client models enable a more comprehensive evaluation. However, when the $m$ value increases from 5 to 10, model performance does not necessarily improve, as incorporating too many client models may introduce redundant or noisy features that undermine exemplar selection. This observation also suggests a promising avenue for future research.

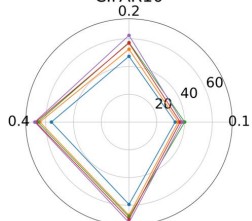 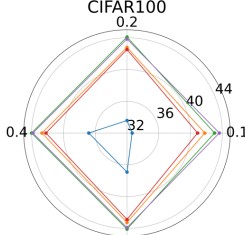 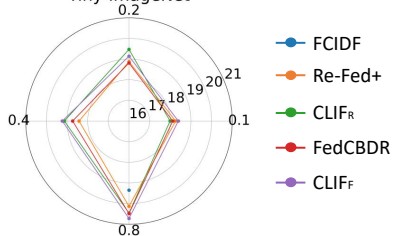

Figure 3: Evaluation of model performance under different client participation ratios across three datasets. Both variants of CLIF outperform the corresponding baselines.

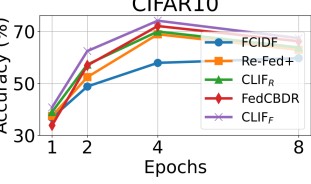 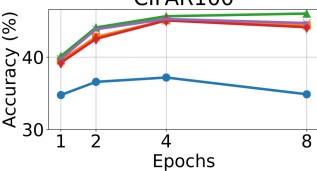 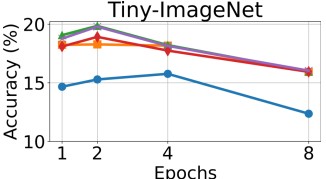

Figure 4: Performance comparison of different methods under varying local training epochs. Our method brings performance gains to the baselines under all settings.

**Fraction of Clients Participating per Round.** This experiment investigates the impact of client participation ratio on model performance. We conduct evaluations on CIFAR10 and CIFAR100 under participation ratios of 0.1, 0.2, 0.4, and 0.8. Benefiting from the increase in client participation ratio, the global distribution becomes more balanced, leading to performance improvements for all methods. More importantly, our approach consistently outperforms the compared baselines, as shown in Figure 3, indicating the effectiveness of cross-view selection in leveraging heterogeneous client data for robust knowledge retention.

**Local Training Epochs.** We adjust the local training epochs from $\{1, 2, 4, 8\}$ for several exemplar replay-based methods. As illustrated in Figure 4, all methods show performance improvements on both datasets as the number of local epochs increases, but the gains become marginal or even decline when the epochs are large (e.g., 8), suggesting that excessive local training may cause overfitting or client drift. Moreover, CLIF$_R$ and CLIF$_F$ consistently outperform their corresponding baselines across different local epochs, validating the effectiveness of the cross-view strategy.

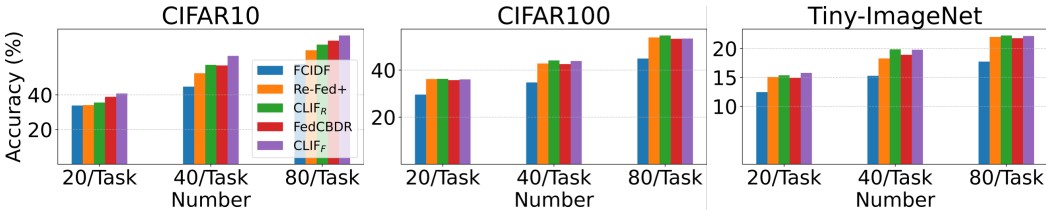

Figure 5: Impact of per-task memory buffer size on model performance across three datasets, where the `CLIF` variants consistently outperform baselines.

**Memory Buffer Size.** Figure 5 shows the performance of different methods under varying memory buffer sizes. For CIFAR10 and CIFAR100, we adjust the number of replay samples per task to 20, 40, and 80. Clearly, all methods benefit from larger buffer sizes, and the `CLIF` variants consistently outperform baselines, demonstrating robustness under limited memory and adaptability to both simple and complex tasks. Moreover, applying cross-view strategy locally enables $\text{CLIF}_R$ to even surpass the original global feature-sharing approach FedCBDR in some cases.

Table 4: Client sample selection / generator training time and peak GPU memory with 20 clients and a ResNet18 backbone.

| Method | CIFAR10 | | CIFAR100 | | Tiny-ImageNet | |
|---|---|---|---|---|---|---|
| | Time (s) | GPU Mem (MB) | Time (s) | GPU Mem (MB) | Time (s) | GPU Mem (MB) |
| LANDER | 27.378 | 3604 | 28.324 | 3604 | 29.748 | 3987 |
| Re-Fed+ | 6.352 | 566 | 6.122 | 566 | 10.966 | 673 |
| FedCBDR | 0.854 | 682 | 0.824 | 682 | 0.882 | 736 |
| $\text{CLIF}_R$ (2-views) | 2.251 | 1449 | 2.155 | 1449 | 4.033 | 1546 |
| $\text{CLIF}_R$ (5-views) | 3.781 | 1449 | 4.206 | 1449 | 7.711 | 1546 |
| $\text{CLIF}_R$ (10-views) | 6.825 | 1449 | 6.580 | 1449 | 14.207 | 1546 |
| $\text{CLIF}_F$F (2-views) | 1.144 | 1511 | 1.180 | 1511 | 1.801 | 1624 |
| $\text{CLIF}_F$F (5-views) | 1.741 | 1511 | 1.846 | 1511 | 2.881 | 1624 |
| $\text{CLIF}_F$F (10-views) | 2.528 | 1511 | 2.604 | 1511 | 4.442 | 1624 |

### 4.5 EVALUATION OF THE MEMORY AND COMPUTATIONAL OVERHEAD

This section added a dedicated analysis of the cost of computing Lewis weights and cross-view fusion. Specifically, following Tables now report the peak GPU memory usage and average client sample selection or generator training time for our method versus Re-Fed+ and FedCBDR, under the same hardware and hyper-parameters, showing that the additional overhead is quantitatively modest. Notably, the GPU peak is dominated by the fixed costs, the main model and all clients' local weights, and the k views run sequentially with their features immediately moved to CPU, larger k only increases runtime, not simultaneous GPU usage. Therefore, $k = 2, 5, 10$ yields nearly identical GPU peak memory.

## 5 CONCLUSIONS AND FUTURE WORK

To address the challenge of balancing effectiveness and privacy in exemplar replay-based FCIL, this paper proposes `CLIF`, which fuses Lewis scores from multiple feature perspectives to achieve consistent importance estimation. Furthermore, it introduces a frequency-based weighted training strategy, which adjusts the loss contribution of samples based on their selection frequency. Experiments on three datasets demonstrate that `CLIF` consistently improves baseline methods by 1%–6%.

Despite `CLIF` performs well, several limitations remain to be explored. Specifically, we plan to design cross-view model evaluation methods to handle undertrained local models and enhance the robustness of data selection in `CLIF`. Moreover, to overcome the issue of replay-induced heterogeneity, exploring multi-source model aggregation techniques is a promising direction for FCIL.

ETHICS STATEMENT

This study involves neither human participants nor animal subjects and does not use sensitive personal data or pursue potentially harmful applications. Experiments were conducted exclusively on publicly available, academically licensed benchmark datasets. The statement on large language model usage is provided in Appendix A.

REPRODUCIBILITY

To ensure reproducibility, we provide an algorithmic outline in Appendix **??** and include the implementation code in the supplementary material.

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

## A    STATEMENT ON THE USE OF LARGE LANGUAGE MODELS

During the preparation of this work, large language models were employed to assist with language polishing. The core research ideas, methodology, experimental results, and conclusions were solely developed by the authors.

## B    PROOF OF THEOREM 1

**Theorem 1.** *Fix a client $k$ and task $t$. For each view $m \in [M]$, let $w_i(A_k^{(m)})$ be the $\ell_p$ Lewis weight of row $i$ and define the fused scores $\tilde{w}_{k,i} = \max_{m \in [M]} w_i(A_k^{(m)})$. Let $T_k = \sum_{i=1}^{n_k} \tilde{w}_{k,i}$. Build a reweighted sampling matrix $S_k$ by drawing $m_k$ rows i.i.d. with probabilities $p_{k,i} = \tilde{w}_{k,i}/T_k$ and per-row weight $(m_k p_{k,i})^{-1/p}$. For any activation functions $f$ that is L-Lipschitz with $f(0) = 0$ and $p \geq 1$. Consider the constrained, reweighted objective on $\mathcal{D}_{k,train}^{(t)}$ for each view $m$:*

$$\tilde{\theta}_k^{(m)} \in \arg\min_{\theta \in E_k^{(m)}} \left\| S_k f(A_k^{(m)}\theta) - S_k y_k \right\|_p^p, \qquad E_k^{(m)} = \left\{ \theta : \|S_k A_k^{(m)}\theta\|_p^p \leq \frac{\|S_k y_k\|_p^p}{\varepsilon L^p} \right\}. \quad (12)$$

*If $m_k \gtrsim_p \varepsilon^{-4} T_k \, d^{\max\{\frac{p}{2}-1,0\}} \log^2 d \cdot \log\left(\frac{dT_k}{\varepsilon}\right)$, then, $\forall m \in [M]$, the following holds with probability at least $0.9$,*

$$\left\| f(A_k^{(m)}\tilde{\theta}_k^{(m)}) - y_k \right\|_p^p \leq C_p \left( \left\| f(A_k^{(m)}\theta_{k,*}^{(m)}) - y_k \right\|_p^p + \varepsilon L^p \|A_k^{(m)}\theta_{k,*}^{(m)}\|_p^p \right), \quad (13)$$

*where $\theta_{k,*}^{(m)} \in \arg\min_\theta \|f(A_k^{(m)}\theta) - y_k\|_p^p$ and $C_p > 0$ is a constant depending only on $p$.*

To prove Theorem 1, we introduce the following two lemmas:

**Lemma 2** (Theorem 3.5 in (Huang et al., 2024)). *Let $p \geq 1$, $f$ be L-Lipschitz with $f(0) = 0$, $A \in \mathbb{R}^{n \times d}$, $y \in \mathbb{R}^n$. Consider a reweighted sampling matrix $S$ built from weights $t_i$ with sampling probabilities $p_i = t_i/m$ where $m = \sum_i t_i$. If $t_i \geq \beta w_i(A)$ for all $i$ with*

$$\beta \gtrsim_p \varepsilon^{-4} d^{\max\{\frac{p}{2}-1,0\}} \log^2 d \cdot \log\left(\sum_{i=1}^n t_i\right), \quad (14)$$

*then, for*

$$\tilde{\theta} \in \arg\min_{\theta \in E} \|Sf(A\theta) - Sy\|_p^p \quad \text{with} \quad E = \left\{ \theta : \|SA\theta\|_p^p \leq \frac{\|Sy\|_p^p}{\varepsilon L^p} \right\}, \quad (15)$$

*it holds with probability at least $0.9$ that*

$$\|f(A\tilde{\theta}) - y\|_p^p \leq C_p \left( \|f(A\theta_*) - y\|_p^p + \varepsilon L^p \|A\theta_*\|_p^p \right), \quad \theta_* = \arg\min_\theta \|f(A\theta) - y\|_p^p. \quad () \; (16)$$

*Here $C_p > 0$ depends only on $p$.*

**Lemma 3** (Corollary 3.6 in (Huang et al., 2024)). *Let $A^{(1)}, \ldots, A^{(M)} \in \mathbb{R}^{n \times d}$ and set $T = \sum_{i=1}^n \max_{m \in [M]} w_i(A^{(m)})$. There exists a shared reweighted sampling matrix $S$ built with sampling probabilities proportional to $\max_m w_i(A^{(m)})$ such that, if*

$$m \gtrsim_p \varepsilon^{-4} T \, d^{\max\{\frac{p}{2}-1,0\}} \log^2 d \cdot \log\left(\frac{dT}{\varepsilon}\right), \quad (17)$$

*then simultaneously for all $m \in [M]$, the conclusion of Lemma 2 holds with $A \leftarrow A^{(m)}$, the same $S$, and the same constraint set $E$ (built with $S$ and $A^{(m)}$).*

Now we provide the proof of Theorem 1.

*Proof.* Apply Lemma 3 with the collection $\{A^{(m)}\}_{m=1}^M = \{A_k^{(m)}\}_{m=1}^M$ and with the target vector $y = y_k$. Our construction of $S_k$ uses sampling probabilities $p_{k,i} \propto \max_m w_i(A_k^{(m)})$ and per–row weights $(m_k p_{k,i})^{-1/p}$, hence $S_k$ matches the $S$ required by Lemma 3.

If the sample size satisfies

$$m_k \gtrsim_p \varepsilon^{-4} T_k d^{\max\{\frac{p}{2}-1,0\}} \log^2 d \cdot \log\left(\frac{dT_k}{\varepsilon}\right),$$

then Lemma 3 guarantees that, with probability at least 0.9, for every view $m \in [M]$, the solution

$$\tilde{\theta}_k^{(m)} \in \arg\min_{\theta \in E_k^{(m)}} \left\| S_k f(A_k^{(m)}\theta) - S_k y_k \right\|_p^p, \qquad E_k^{(m)} = \left\{ \theta : \|S_k A_k^{(m)}\theta\|_p^p \leq \frac{\|S_k y_k\|_p^p}{\varepsilon L^p} \right\}, \quad (18)$$

satisfies

$$\left\| f(A_k^{(m)}\tilde{\theta}_k^{(m)}) - y_k \right\|_p^p \leq C_p\left( \left\| f(A_k^{(m)}\theta_{k,*}^{(m)}) - y_k \right\|_p^p + \varepsilon L^p \|A_k^{(m)}\theta_{k,*}^{(m)}\|_p^p \right), \quad (19)$$

where $\theta_{k,*}^{(m)} \in \arg\min_\theta \|f(A_k^{(m)}\theta) - y_k\|_p^p$.

Under the stated sample complexity on $m_k$ and the Lewis-weight–max fusion sampling, the desired inequality holds for all views simultaneously with probability at least 0.9, completing the proof. □

## C  DEFINITION OF RECONSTRUCTION ERROR

We further assess the representativeness of the subset by evaluating how well its induced subspace can approximate the full dataset. Specifically, let $U_r \in \mathbb{R}^{d \times r}$ denote the top-$r$ eigenvectors of the subset covariance $C_S = A_S^\top A_S$. The reconstruction of the global feature matrix is given by

$$\hat{A} = (A - \mu)U_r U_r^\top, \quad (20)$$

where $\mu$ is the centering mean (either global or subset mean). The *reconstruction error (RE)* is then defined as

$$\text{RE}(A, A_S, r) = \frac{\|(A - \mu) - (A - \mu)U_r U_r^\top\|_F^2}{\|A - \mu\|_F^2}. \quad (21)$$

**A lower RE indicates that the subset preserves the global geometry more effectively.**

## D  OTHER RELATED WORKS

### D.1  REGULARIZATION-BASED METHODS

Beyond replay-oriented techniques, a complementary research direction lies in regularization-based methods, which constrain model updates to preserve knowledge from previously learned tasks. These methods can be broadly divided into two subgroups. Parameter-level regularization restricts parameter updates by estimating their importance, with representative techniques such as elastic weight consolidation that can be seamlessly adapted to the federated setting (Kirkpatrick et al., 2017; Iqbal et al., 2025). Output-level regularization, on the other hand, aims to stabilize representations and outputs across tasks, typically through consistency regularization, prototype alignment, or logit matching (Yoo & Park, 2024; Psaltis et al., 2023; Li & Hoiem, 2017; Tan et al., 2024; Feng et al., 2025). While these approaches reduce reliance on replay and lower buffer demands, their effectiveness often deteriorates as the number of tasks increases.

## E  OTHER RESULTS

### E.1  PERFORMANCE COMPARISON ON DOMAINNET DATASET

This section added a new experimental section on DomainNet, containing roughly 600K images. Concretely, we first split DomainNet into training and test sets with an 8:2 ratio for each category. Following standard practice for federated non-IID partitioning, we then distribute the training data to 20 clients using a Dirichlet prior over label distributions with concentration parameters $\beta = \{0.1, 0.5\}$. On top of this client partition, we adopt a 10-task incremental split, and each client is allowed to replay 340 samples per task in our continual federated setting. We evaluate both a ResNet-18 and CLIP (ViT-Base/16) to further validate generalizability across model capacities,

Table 5: Results on DomainNet with 20 clients.

| 20 Clients | DomainNet | | | |
| | ResNet18 | | CLIP | |
| | $\beta = 0.1$ | $\beta = 0.5$ | $\beta = 0.1$ | $\beta = 0.5$ |
|---|---|---|---|---|
| Re-Fed+ | 14.82 | 46.82 | 55.35 | 57.63 |
| $\text{CLIF}_R$ | 16.21 | 48.64 | 56.74 | 60.03 |
| FedCBDR | 15.28 | 46.73 | 55.47 | 58.80 |
| $\text{CLIF}_F$ | 16.33 | 50.70 | 57.01 | 61.12 |

where CLIP is trained by incorporating multimodal prompt tuning, the size of the learnable textual prompt and visual prompt are 10*512 and 10*768, respectively.

As reported in the Table 5, our method consistently outperforms all baselines on DomainNet across both Dirichlet settings and both backbones, achieving higher average accuracy while reducing forgetting under substantially more challenging non-IID conditions.

## E.2 PERFORMANCE EVALUATION UNDER A LARGER NUMBER OF CLIENTS AND MORE EXTREME HETEROGENEITY SCENARIOS

Table 6: Performance comparison under $\beta = 0.1$ with 100 clients.

| $\beta = 0.1$ | CIFAR10 | | CIFAR100 | | TinyImageNet | |
| | ResNet18 | CLIP | ResNet18 | CLIP | ResNet18 | CLIP |
|---|---|---|---|---|---|---|
| Re-Fed+ | 35.59 | 90.02 | 26.46 | 68.51 | 18.78 | 34.38 |
| $\text{CLIF}_R$ | 39.75 | 90.86 | 31.65 | 70.22 | 22.80 | 39.27 |
| FedCBDR | 38.82 | 86.74 | 28.15 | 70.18 | 18.45 | 36.92 |
| $\text{CLIF}_F$F | 40.11 | 90.53 | 32.17 | 70.49 | 22.72 | 41.01 |

This section extends our experiments in two directions. First, we evaluate the proposed CLIF framework with both ResNet-18 and CLIP (ViT-B/16) backbones on 100 clients, and compare it against the same state-of-the-art baselines as in the main text. Second, we investigate more extreme data heterogeneity by using a smaller Dirichlet parameter ($\beta = 0.1$) when partitioning data across clients. The results are summarized in the Table 6, which shows that CLIF consistently outperforms the baselines under 100-client settings and remains robust under highly non-IID partitions, indicating good scalability and robustness of our method in more challenging FL regimes.

## E.3 PERFORMANCE COMPARISON WITH CLIP BACKBONE

This section additionally incorporated a CLIP backbone to better reflect modern practice. We evaluate CLIF with this CLIP backbone on CIFAR-10/100, TinyImageNet, and DomainNet under various heterogeneity levels ($\beta = 0.1, 0.5$) and 20-client settings. The results reported in the Table 7 show that CLIF continues to outperform strong baselines when built on top of a high-quality, pre-trained transformer backbone, indicating that the method is effective not only when training a small ResNet-18 from scratch, but also in the more realistic pre-training / fine-tuning regime.

## E.4 PERFORMANCE EVALUATION OF VARIOUS METHODS AT EACH INCREMENTAL STAGE

In this section, we evaluate all methods on incremental tasks across three datasets. As illustrated in Figure 6, the relative improvement of CLIF variants over baselines is more pronounced on CIFAR10, but remains visible on CIFAR100 and Tiny-ImageNet. The superior performance of $\text{CLIF}_F$ and $\text{CLIF}_R$ can be attributed to stronger representative sample selection and the contribution weighting of critical samples, which help mitigate forgetting and improve knowledge transfer. In particular,

Table 7: Average accuracy (%) on CIFAR10, CIFAR100, and TinyImageNet under different task splits and heterogeneity levels.

| Method | CIFAR10 | | CIFAR100 | | TinyImageNet | |
|---|---|---|---|---|---|---|
| | $\beta = 0.1$ | $\beta = 0.5$ | $\beta = 0.1$ | $\beta = 0.5$ | $\beta = 0.1$ | $\beta = 0.5$ |
| **Task 3/5/5** | | | | | | |
| *iCarl* | 87.35 | 90.57 | 59.38 | 63.48 | 16.94 | 22.08 |
| Re-Fed+ | 89.02 | 90.68 | 71.96 | 72.35 | 31.29 | 47.58 |
| $\text{CLIF}_R$ | 92.48 | 93.34 | 73.47 | 73.98 | 37.45 | 48.83 |
| FedCBDR | 90.79 | 92.33 | 73.07 | 74.07 | 35.26 | 46.30 |
| $\text{CLIF}_F$F | 92.85 | 92.89 | 73.09 | 74.28 | 35.84 | 48.19 |
| **Task 5/10/10** | | | | | | |
| *iCarl* | 88.67 | 90.74 | 63.48 | 65.39 | 17.28 | 20.16 |
| Re-Fed+ | 91.19 | 92.04 | 72.46 | 72.60 | 35.33 | 41.87 |
| $\text{CLIF}_R$ | 92.54 | 93.51 | 73.88 | 74.03 | 38.50 | 45.69 |
| FedCBDR | 91.57 | 91.86 | 72.57 | 73.01 | 36.57 | 45.44 |
| $\text{CLIF}_F$F | 93.11 | 93.18 | 73.60 | 74.94 | 38.85 | 46.92 |

on CIFAR10, $\text{CLIF}_F$ achieves the most stable performance, maintaining higher accuracy across later tasks compared to $\text{CLIF}_R$.

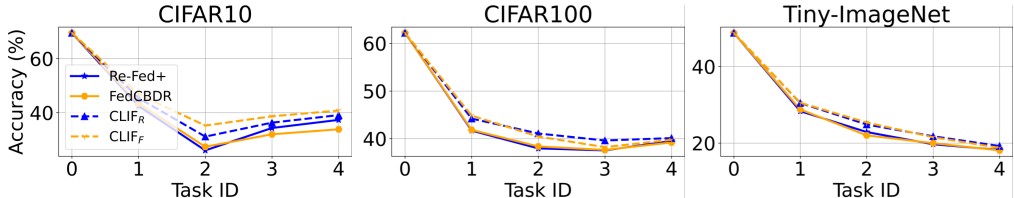

Figure 6: Incremental performance evaluation on CIFAR-10, CIFAR-100, and Tiny-ImageNet with 20 clients under heterogeneity $\beta = 0.5$, a 5-task splitting, client participation rate $p = 0.2$, and local training epoch $e = 1$. $\text{CLIF}_F$ and $\text{CLIF}_R$ consistently outperform their baselines on all cases.

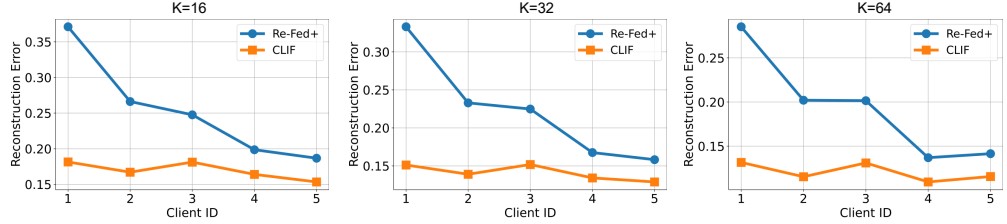

Figure 7: Reconstruction error comparison between Re-Fed+ and $\text{CLIF}$ under different subspace dimensions $K = \{16, 32, 64\}$. Across all values of $K$, $\text{CLIF}$ consistently achieves lower reconstruction error, indicating superior preservation of the global feature space geometry.

### E.5 Evaluating Geometric Structure Preservation

In this section, we evaluate the representativeness of replay subsets using the reconstruction error (RE) metric, which measures how well the subspace spanned by the selected samples can approximate the global feature space. Specifically, for each client, we construct a low-dimensional subspace of dimension $K \in \{16, 32, 64\}$ from the selected subsets and compute the reconstruction error of the full client data. The results are reported in Figure 7. It is evident that across all values of $K$, our proposed $\text{CLIF}$ consistently achieves lower reconstruction error compared with the baseline Re-

Table 8: Average set overlap (Jaccard index) and average ranking consistency (Spearman correlation) between the exemplars selected (Top-40) at the end of task 1 and those re-selected after the backbone has continuously evolved during task 2 on all clients.

| | CIFAR10 | | | CIFAR100 | | |
|---|---|---|---|---|---|---|
| | 2-views | 5-views | 10-views | 2-views | 5-views | 10-views |
| **Jaccard** | 0.6127 | 0.6038 | 0.6072 | 0.5723 | 0.5842 | 0.5693 |
| **Spearman** | 0.8366 | 0.8147 | 0.8275 | 0.8283 | 0.8416 | 0.8154 |

Table 9: Total communication over $n$ rounds. Here $|\theta|$: full model size; $|\theta_{\text{enc}}|$: encoder size; $n$: number of rounds a client participates; $k$: number of additional views; $|D|$: number of uploaded samples; $d$: feature dimension; $|B|$: replay buffer size.

| Method | Total Download | Total Upload | Notes |
|---|---|---|---|
| Re-Fed+ (Local replay) | $n\,|\theta|$ | $n\,|\theta|$ | Replay selection is purely local; no extra transfers beyond baseline. |
| $\text{CLIF}_R$ (Cross-view replay) | $n\,|\theta| + k\,|\theta_{enc}|$ | $n\,|\theta|$ | At the final round, an extra download of $k$ client encoders $\theta_{enc}$ is performed; uplink remains the same as baseline. |
| FedCBDR (Global replay) | $n\,|\theta| + |B|$ | $n\,|\theta| + |D|\,d$ | Upload $|D|$ feature vectors of dimension $d$ once at selection (extra uplink $= |D|\,d$). The server identifies the sample IDs and sends them back to the respective clients. |
| $\text{CLIF}_F$ (Global cross-view replay) | $n\,|\theta| + k\,|\theta_{enc}| + |B|$ | $n\,|\theta| + k\,|D|\,d$ | Compared with FedCBDR, it incurs an extra download of $k$ encoders and an extra upload of features extracted by $k$ models. |

Fed+, demonstrating that the multi-view Lewis weight fusion strategy preserves the global geometry of the feature space more effectively.

### E.6 STABILITY OF CROSS-TASK SAMPLE SELECTION

This section measures the average set overlap (Jaccard index) and average ranking consistency (Spearman correlation) between the exemplars selected (Top-40) at the end of task 1 and those re-selected after the backbone has continuously evolved during task 2 on client 1. As shown in Table 8, even under configurations with up to 100 rounds of federated training, we still observe that the overlap remains consistently high, indicating that the exemplar set remains stable despite substantial model updates. These values also indicate that, when selecting Top-40 exemplars, about 30 samples are shared between the exemplar sets across model snapshots. Specifically, a Jaccard score of 0.6000 corresponds to 30 overlapping samples, 0.5686 corresponds to 29.

### E.7 COMMUNICATION AND BANDWIDTH OVERHEAD

All methods inherit the baseline FL cost of downloading and uploading one full model per participating round, i.e., $n|\theta|$ download and $n|\theta|$ upload, as shown in Table 9, where $|\theta|$ is full model size; $|\theta_{\text{enc}}|$ denotes encoder size; $n$ is the number of rounds a client participates; $k$ is the number of

Table 10: Analysis of different view construction strategies on model performance across three datasets under varying levels of heterogeneity ($\beta = \{0.5, 1.0\}$). Two task-split settings are adopted: 3/5/5 and 5/10/10, where CIFAR10, CIFAR100, and Tiny-ImageNet are split into 3/5, 5/10, and 5/10 tasks, respectively. Results are averaged over three random seeds, and we report mean±standard deviation. The best results are highlighted in **bold**.


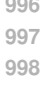

| | CIFAR10 | | CIFAR100 | | Tiny-ImageNet | | CIFAR10 | | CIFAR100 | | Tiny-ImageNet | |
|---|---|---|---|---|---|---|---|---|---|---|---|---|
| | **20 Clients (3/5/5 Task-Splits)** | | | | | | **40 Clients (3/5/5 Task-Splits)** | | | | | |
| | $\beta=0.5$ | $\beta=1.0$ | $\beta=0.5$ | $\beta=1.0$ | $\beta=0.5$ | $\beta=1.0$ | $\beta=0.5$ | $\beta=1.0$ | $\beta=0.5$ | $\beta=1.0$ | $\beta=0.5$ | $\beta=1.0$ |
| Re-Fed+ | $52.45_{\pm1.9}$ | $61.74_{\pm1.1}$ | $42.79_{\pm2.3}$ | $43.56_{\pm1.4}$ | $18.88_{\pm2.0}$ | $19.46_{\pm1.6}$ | $67.16_{\pm2.5}$ | $69.42_{\pm1.3}$ | $45.12_{\pm1.8}$ | $46.04_{\pm0.9}$ | $18.28_{\pm2.7}$ | $18.89_{\pm1.8}$ |
| CLIF$_R$ | $57.30_{\pm1.2}$ | $64.29_{\pm1.3}$ | $44.10_{\pm1.1}$ | $45.83_{\pm1.6}$ | $\mathbf{19.47_{\pm0.6}}$ | $\mathbf{19.96_{\pm1.8}}$ | $69.44_{\pm1.2}$ | $72.10_{\pm2.1}$ | $\mathbf{46.65_{\pm1.2}}$ | $46.40_{\pm1.3}$ | $\mathbf{19.86_{\pm1.1}}$ | $20.27_{\pm1.5}$ |
| CLIF$_{R+G}$ | $57.13_{\pm1.5}$ | $64.57_{\pm1.1}$ | $43.86_{\pm0.9}$ | $\mathbf{45.92_{\pm1.4}}$ | $19.21_{\pm1.1}$ | $19.58_{\pm1.2}$ | $\mathbf{69.79_{\pm1.1}}$ | $\mathbf{72.31_{\pm1.4}}$ | $46.24_{\pm1.6}$ | $46.31_{\pm1.1}$ | $19.52_{\pm1.8}$ | $20.11_{\pm1.2}$ |
| CLIF$_{R+B}$ | $\mathbf{57.77_{\pm1.3}}$ | $\mathbf{64.61_{\pm1.7}}$ | $\mathbf{44.31_{\pm1.8}}$ | $45.16_{\pm1.2}$ | $19.41_{\pm0.7}$ | $19.73_{\pm1.4}$ | $69.69_{\pm1.1}$ | $72.16_{\pm1.3}$ | $46.41_{\pm1.6}$ | $\mathbf{46.72_{\pm1.5}}$ | $19.39_{\pm1.0}$ | $\mathbf{20.44_{\pm1.4}}$ |
| FedCBDR | $56.94_{\pm1.6}$ | $66.01_{\pm2.2}$ | $42.52_{\pm1.5}$ | $44.85_{\pm0.8}$ | $18.81_{\pm1.9}$ | $19.79_{\pm1.2}$ | $70.65_{\pm2.3}$ | $72.57_{\pm1.6}$ | $45.74_{\pm2.4}$ | $46.20_{\pm1.1}$ | $18.92_{\pm2.3}$ | $19.24_{\pm2.1}$ |
| CLIF$_F$ | $\mathbf{62.47_{\pm2.4}}$ | $68.29_{\pm1.0}$ | $\mathbf{43.84_{\pm1.7}}$ | $46.34_{\pm1.2}$ | $19.14_{\pm0.9}$ | $19.60_{\pm1.8}$ | $73.23_{\pm1.5}$ | $\mathbf{76.21_{\pm2.0}}$ | $\mathbf{46.86_{\pm2.1}}$ | $47.26_{\pm0.7}$ | $\mathbf{19.79_{\pm1.4}}$ | $\mathbf{19.99_{\pm1.3}}$ |
| CLIF$_{F+G}$ | $61.89_{\pm1.7}$ | $\mathbf{68.52_{\pm1.2}}$ | $43.41_{\pm1.3}$ | $\mathbf{46.77_{\pm1.6}}$ | $18.92_{\pm1.1}$ | $19.82_{\pm1.2}$ | $72.83_{\pm1.5}$ | $75.53_{\pm1.4}$ | $46.69_{\pm1.1}$ | $\mathbf{47.71_{\pm1.6}}$ | $19.55_{\pm1.5}$ | $19.38_{\pm1.3}$ |
| CLIF$_{F+B}$ | $62.07_{\pm1.6}$ | $68.44_{\pm1.5}$ | $43.51_{\pm1.2}$ | $46.52_{\pm1.5}$ | $\mathbf{19.22_{\pm0.8}}$ | $\mathbf{19.91_{\pm1.5}}$ | $\mathbf{73.41_{\pm1.3}}$ | $75.88_{\pm1.6}$ | $46.15_{\pm1.8}$ | $46.86_{\pm1.2}$ | $19.21_{\pm1.9}$ | $19.57_{\pm1.4}$ |

| | CIFAR10 | | CIFAR100 | | Tiny-ImageNet | | CIFAR10 | | CIFAR100 | | Tiny-ImageNet | |
|---|---|---|---|---|---|---|---|---|---|---|---|---|
| | **20 Clients (5/10/10 Task-Splits)** | | | | | | **40 Clients (5/10/10 Task-Splits)** | | | | | |
| | $\beta=0.5$ | $\beta=1.0$ | $\beta=0.5$ | $\beta=1.0$ | $\beta=0.5$ | $\beta=1.0$ | $\beta=0.5$ | $\beta=1.0$ | $\beta=0.5$ | $\beta=1.0$ | $\beta=0.5$ | $\beta=1.0$ |
| Re-Fed+ | $49.10_{\pm1.7}$ | $60.32_{\pm1.4}$ | $39.18_{\pm0.9}$ | $40.42_{\pm1.3}$ | $15.19_{\pm1.8}$ | $16.45_{\pm1.1}$ | $61.92_{\pm1.2}$ | $68.28_{\pm1.6}$ | $52.15_{\pm1.4}$ | $53.20_{\pm1.7}$ | $16.26_{\pm0.6}$ | $16.68_{\pm1.5}$ |
| CLIF$_R$ | $52.34_{\pm0.8}$ | $\mathbf{63.04_{\pm1.2}}$ | $40.23_{\pm1.5}$ | $41.05_{\pm1.9}$ | $16.27_{\pm1.0}$ | $17.51_{\pm1.6}$ | $63.44_{\pm1.7}$ | $\mathbf{71.34_{\pm1.1}}$ | $53.26_{\pm1.8}$ | $\mathbf{53.88_{\pm0.7}}$ | $18.04_{\pm2.1}$ | $\mathbf{18.32_{\pm1.3}}$ |
| CLIF$_{R+G}$ | $51.88_{\pm1.4}$ | $62.54_{\pm1.1}$ | $40.34_{\pm1.2}$ | $40.85_{\pm2.1}$ | $\mathbf{16.31_{\pm1.7}}$ | $\mathbf{17.73_{\pm1.2}}$ | $62.81_{\pm1.5}$ | $70.76_{\pm1.2}$ | $53.44_{\pm1.4}$ | $53.84_{\pm1.1}$ | $17.75_{\pm1.7}$ | $18.06_{\pm1.6}$ |
| CLIF$_{R+B}$ | $\mathbf{52.61_{\pm1.1}}$ | $62.41_{\pm1.5}$ | $\mathbf{40.45_{\pm1.5}}$ | $\mathbf{41.52_{\pm1.5}}$ | $16.18_{\pm1.3}$ | $17.60_{\pm1.1}$ | $\mathbf{63.52_{\pm1.6}}$ | $71.22_{\pm1.5}$ | $\mathbf{53.56_{\pm1.6}}$ | $53.41_{\pm1.7}$ | $\mathbf{18.37_{\pm2.0}}$ | $18.13_{\pm1.3}$ |
| FedCBDR | $50.50_{\pm1.6}$ | $65.47_{\pm0.9}$ | $39.37_{\pm1.3}$ | $39.52_{\pm1.7}$ | $15.38_{\pm1.2}$ | $17.40_{\pm1.8}$ | $66.78_{\pm1.4}$ | $74.05_{\pm1.0}$ | $51.09_{\pm1.9}$ | $52.30_{\pm1.1}$ | $16.08_{\pm0.7}$ | $16.11_{\pm1.5}$ |
| CLIF$_F$ | $52.85_{\pm1.2}$ | $68.14_{\pm1.9}$ | $40.42_{\pm1.1}$ | $41.39_{\pm1.4}$ | $16.49_{\pm1.8}$ | $17.63_{\pm1.5}$ | $\mathbf{68.06_{\pm0.9}}$ | $75.54_{\pm1.3}$ | $\mathbf{53.30_{\pm1.6}}$ | $\mathbf{54.52_{\pm1.0}}$ | $17.14_{\pm1.3}$ | $17.18_{\pm1.7}$ |
| CLIF$_{F+G}$ | $52.24_{\pm1.4}$ | $67.65_{\pm1.6}$ | $40.61_{\pm1.3}$ | $41.24_{\pm1.7}$ | $\mathbf{16.63_{\pm1.2}}$ | $\mathbf{17.84_{\pm1.1}}$ | $67.67_{\pm1.5}$ | $75.31_{\pm1.4}$ | $52.83_{\pm1.1}$ | $54.01_{\pm1.6}$ | $\mathbf{17.43_{\pm1.2}}$ | $16.83_{\pm1.4}$ |
| CLIF$_{F+B}$ | $\mathbf{53.11_{\pm1.8}}$ | $\mathbf{68.43_{\pm1.2}}$ | $\mathbf{40.77_{\pm1.0}}$ | $41.51_{\pm1.5}$ | $16.31_{\pm1.3}$ | $17.73_{\pm1.6}$ | $67.44_{\pm1.1}$ | $\mathbf{75.67_{\pm1.7}}$ | $53.11_{\pm1.3}$ | $54.26_{\pm0.8}$ | $17.28_{\pm1.6}$ | $\mathbf{17.41_{\pm1.3}}$ |

additional views; $|D|$ is the number of uploaded samples; $d$ is feature dimension; $|B|$ denotes the replay buffer size.

*Re-Fed+ (Local replay).* Replay selection is purely local; the total remains $n|\theta|$ download and $n|\theta|$ upload.

CLIF$_R$ *(Cross-view replay).* A one-off multi-view step at the final round dispatches $k$ *encoders* to each client. Thus the total becomes $n|\theta| + k|\theta_{\text{enc}}|$ download and $n|\theta|$ upload (uplink unchanged from baseline).

*FedCBDR (Global replay).* Each client downloads a list of $|B|$ sample IDs (not the samples themselves) once and uploads the features of $|D|$ samples (feature dimension $d$). Totals: $n|\theta| + |B|$ download and $n|\theta| + |D|d$ upload.

CLIF$_F$ *(Global cross-view replay).* This combines FedCBDR with the multi-view encoders: totals are $n|\theta| + k|\theta_{\text{enc}}| + |B|$ download and $n|\theta| + k|D|d$ upload. Equivalently, relative to FedCBDR it adds downloading $k$ encoders and uploading features extracted by $k$ models.

### E.8 ANALYSIS OF THE IMPACT OF DIFFERENT VIEW CONSTRUCTION STRATEGIES ON MODEL PERFORMANCE

To further mitigate the risk of privacy leakage from sharing client models, we propose two alternative strategies:

1. **Gaussian perturbation** ($G$) Truex et al. (2020); Wei et al. (2023): add small Gaussian noise to the client parameters

$$\tilde{\theta}_k = \theta_k + \epsilon, \quad \epsilon \sim \mathcal{N}(0, \sigma^2 I), \tag{22}$$

where $\sigma^2 = 0.1$ controls the noise variance.

Table 11: Attack performance of DLG under the original setting and two variants (+G and +B), measured by PSNR (lower is better).

|  | Ori | +G | +B |
|---|---|---|---|
| **PSNR** | 16.58 | 10.34 | 11.28 |

2. **Dynamic Beta-weighted combination** ($B$) Mao et al. (2020): generate weighting coefficients from a Beta distribution to combine two different model parameters

$$\tilde{\theta} = \eta\,\theta_a + (1 - \eta)\,\theta_b, \quad \eta \sim \text{Beta}(1, 1). \tag{23}$$

After perturbation or combination, the resulting model is then sent to the corresponding client. Moreover, the performance of these alternative approaches is evaluated, with the comparison results presented in Table 10. As observed, their performance is generally similar, indicating that the proposed alternatives are largely equivalent in effectiveness. Furthermore, following the DLG attack protocol Zhu et al. (2019), we perform gradient inversion attacks on three types of parameters (original, with added Gaussian noise, and random combinations). We then compute the PSNR (lower is better) between the reconstructed images and the corresponding original images, and the results are summarized in the Table 11.

### E.9 QUALITATIVE ANALYSIS OF SAMPLE SELECTION

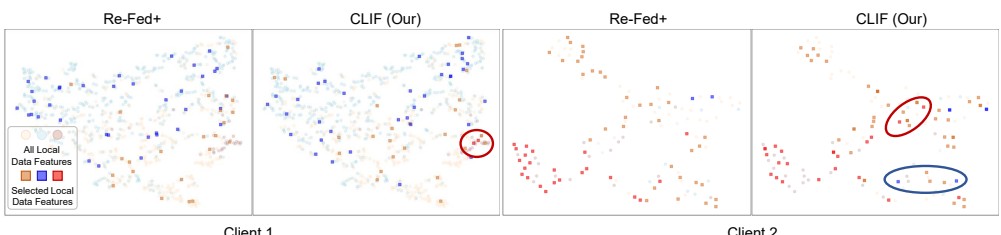

Figure 8: Comparison between CLIF and Re-Fed+ in sample replay. Re-Fed+ tends to lose samples from minority classes and exhibits limited diversity in its selection. In contrast, CLIF is able to retain samples from all classes, while ensuring greater diversity and more representative coverage across the feature space.

To further examine the effectiveness of different replay strategies, we conduct a qualitative comparison between CLIF and Re-Fed+ in terms of sample selection. As shown in Figure 8, Re-Fed+ often fails to preserve minority-class instances, leading to biased replay buffers that emphasize majority classes. Moreover, the selected samples tend to cluster in a few regions of the feature space, indicating limited diversity and insufficient representativeness. In contrast, CLIF consistently maintains coverage across all classes, ensuring that even minority categories are represented in the replay buffer. The selected samples are also more dispersed and complementary to each other, which highlights their diversity and representativeness. Such balanced and globally aware selection is crucial for mitigating forgetting and maintaining stable performance in federated class-incremental learning.

### E.10 PERFORMANCE ANALYSIS UNDER VARIOUS RE-WEIGHTING SCHEMES

This section compares the impact of different re-weighted losses on model performance, including our sampling-frequency re-weighted (FWT) loss, Focal Loss, and a class-balanced re-weighted loss. As reported in Table 12, our method achieves the best performance in most cases, indicating that prioritizing and re-weighting higher-information samples is key to improving model robustness and generalization. In contrast, Focal Loss focuses on "hard" examples, which can over-weight noisy labels and still doesn't address class coverage or replay-buffer bias. Class-balanced ($CB$) loss

Table 12: Analysis of different re-weighting strategies on model performance across three datasets under varying levels of heterogeneity ($\beta = \{0.5, 1.0\}$). Two task-split settings are adopted: 3/5/5 and 5/10/10, where CIFAR10, CIFAR100, and Tiny-ImageNet are split into 3/5, 5/10, and 5/10 tasks, respectively. Results are averaged over three random seeds, and we report mean±standard deviation. The best results are highlighted in **bold**. $Focal$ denotes the Focal loss. $CB$ indicates the class frequency-based re-weighting.

| | CIFAR10 | | CIFAR100 | | Tiny-ImageNet | | CIFAR10 | | CIFAR100 | | Tiny-ImageNet | |
|---|---|---|---|---|---|---|---|---|---|---|---|---|
| | 20 Clients (3/5/5 Task-Splits) | | | | | | 40 Clients (3/5/5 Task-Splits) | | | | | |
| | $\beta=0.5$ | $\beta=1.0$ | $\beta=0.5$ | $\beta=1.0$ | $\beta=0.5$ | $\beta=1.0$ | $\beta=0.5$ | $\beta=1.0$ | $\beta=0.5$ | $\beta=1.0$ | $\beta=0.5$ | $\beta=1.0$ |
| Re-Fed+ | $52.45_{\pm1.9}$ | $61.74_{\pm1.1}$ | $42.79_{\pm2.3}$ | $43.56_{\pm1.4}$ | $18.88_{\pm2.0}$ | $19.46_{\pm1.6}$ | $67.16_{\pm2.5}$ | $69.42_{\pm1.3}$ | $45.12_{\pm1.8}$ | $46.04_{\pm0.9}$ | $18.28_{\pm2.7}$ | $18.89_{\pm1.8}$ |
| CLIF$_R$ | $\mathbf{57.30_{\pm1.2}}$ | $\mathbf{64.29_{\pm1.3}}$ | $44.10_{\pm1.1}$ | $\mathbf{45.83_{\pm1.6}}$ | $\mathbf{19.47_{\pm0.6}}$ | $\mathbf{19.96_{\pm1.8}}$ | $\mathbf{69.44_{\pm1.2}}$ | $\mathbf{72.10_{\pm2.1}}$ | $46.65_{\pm1.2}$ | $\mathbf{46.40_{\pm1.3}}$ | $\mathbf{19.86_{\pm1.1}}$ | $\mathbf{20.27_{\pm1.5}}$ |
| CLIF$_{R+Focal}$ | $54.32_{\pm1.7}$ | $63.21_{\pm1.2}$ | $42.13_{\pm1.4}$ | $43.73_{\pm1.1}$ | $18.44_{\pm1.8}$ | $18.61_{\pm0.9}$ | $68.15_{\pm1.3}$ | $71.38_{\pm1.1}$ | $45.82_{\pm1.5}$ | $45.63_{\pm1.8}$ | $19.32_{\pm2.2}$ | $19.56_{\pm2.1}$ |
| CLIF$_{R+CB}$ | $56.53_{\pm1.6}$ | $63.92_{\pm1.3}$ | $\mathbf{44.51_{\pm1.4}}$ | $44.73_{\pm1.2}$ | $19.26_{\pm1.1}$ | $19.54_{\pm1.6}$ | $68.35_{\pm1.2}$ | $71.70_{\pm1.8}$ | $\mathbf{46.83_{\pm1.4}}$ | $45.78_{\pm1.2}$ | $19.12_{\pm1.1}$ | $19.29_{\pm1.8}$ |
| | 20 Clients (5/10/10 Task-Splits) | | | | | | 40 Clients (5/10/10 Task-Splits) | | | | | |
| | $\beta=0.5$ | $\beta=1.0$ | $\beta=0.5$ | $\beta=1.0$ | $\beta=0.5$ | $\beta=1.0$ | $\beta=0.5$ | $\beta=1.0$ | $\beta=0.5$ | $\beta=1.0$ | $\beta=0.5$ | $\beta=1.0$ |
| Re-Fed+ | $49.10_{\pm1.7}$ | $60.32_{\pm1.4}$ | $39.18_{\pm0.9}$ | $40.42_{\pm1.3}$ | $15.19_{\pm1.8}$ | $16.45_{\pm1.1}$ | $61.92_{\pm1.2}$ | $68.28_{\pm1.6}$ | $52.15_{\pm1.4}$ | $53.20_{\pm1.7}$ | $16.26_{\pm0.6}$ | $16.68_{\pm1.5}$ |
| CLIF$_R$ | $\mathbf{52.34_{\pm0.8}}$ | $\mathbf{63.04_{\pm1.2}}$ | $40.23_{\pm1.5}$ | $41.05_{\pm1.9}$ | $\mathbf{16.27_{\pm1.0}}$ | $\mathbf{17.51_{\pm1.6}}$ | $63.44_{\pm1.7}$ | $\mathbf{71.34_{\pm1.1}}$ | $\mathbf{53.26_{\pm1.8}}$ | $\mathbf{53.88_{\pm0.7}}$ | $18.04_{\pm2.1}$ | $\mathbf{18.32_{\pm1.3}}$ |
| CLIF$_{R+Focal}$ | $51.36_{\pm1.5}$ | $61.52_{\pm1.6}$ | $\mathbf{40.57_{\pm1.3}}$ | $\mathbf{41.11_{\pm1.6}}$ | $15.47_{\pm2.1}$ | $16.68_{\pm1.5}$ | $62.33_{\pm1.6}$ | $69.92_{\pm1.8}$ | $53.18_{\pm1.3}$ | $53.22_{\pm1.4}$ | $\mathbf{18.07_{\pm1.8}}$ | $17.76_{\pm1.3}$ |
| CLIF$_{R+CB}$ | $51.12_{\pm1.4}$ | $62.08_{\pm1.6}$ | $39.92_{\pm1.4}$ | $40.82_{\pm1.7}$ | $15.78_{\pm1.2}$ | $17.42_{\pm1.4}$ | $\mathbf{63.61_{\pm1.3}}$ | $70.64_{\pm1.9}$ | $52.77_{\pm1.3}$ | $53.65_{\pm1.6}$ | $17.67_{\pm1.8}$ | $17.74_{\pm2.0}$ |

reweights only by counts, ignoring sample informativeness and task dynamics, so an unbalanced buffer under federated non-IID remains unfixed.

### E.11 Discussion on Model Selection

This section discusses two representative directions, which we consider as part of future work:

- **Selection based on local validation performance.** Each client could evaluate candidate models on a small local validation set and prioritize those with lower validation loss or higher accuracy. This may better align the selected views with the client's own data distribution, but requires multiple additional forward passes over several external models.

- **Selection based on feature similarity or diversity.** Using a shared probe set or a small subset of local samples, one could compute feature representations for candidate models and choose views according to their similarity to (or diversity from) the local model in feature space. For example, favoring more different models to obtain complementary information. This explicitly exploits feature geometry to construct diverse and complementary views, but needs extra bookkeeping of feature statistics or similarity matrices and may incur additional computation/communication overhead.

Overall, these strategies are promising but involve non-trivial trade-offs between selection quality, computational and communication cost, and system complexity. We will clarify this point in the paper and explicitly highlight more advanced model selection mechanisms as an important direction for future work building on top of our current framework.

