# OpenReview forum: "Cross-View Lewis Weight Fusion Empowering Exemplar Replay for Federated Class-Incremental Learning"
_ICLR.cc/2026/Conference — Submitted to ICLR 2026_

### Official Review · Reviewer_AsPT · 2025-10-29

**Soundness:** 3
**Presentation:** 3
**Contribution:** 3
**Rating:** 6
**Confidence:** 3

**Summary:**

This paper proposes a method, CLIF, for balancing effectiveness and privacy in exemplar replay-based Federated Class-Incremental Learning (FCIL), which uses multi-view importance scores to select representative samples. In order to achieve that, they perform consistent importance estimation by computing and integrating the cross-view Lewis weights to ensure the representativeness of the replay subset consisting of selected samples. Based on this, they depend on the selection frequency of each sample to adjust its contribution to the optimization objective during the frequency-based weighted training. They present a theoretical analysis to guarantee the soundness and effectiveness of CLIF.

**Strengths:**

1. The paper presents a novel method of exemplar relay to mitigate the limitations of effectiveness and rigorous theoretical guarantees in prior works. The idea of Lewis weight fusion across views is original and contributes to the literature on Federated Class-Incremental Learning.
2. The paper demonstrates solid experimental quality, with well-designed comparisons against strong baselines such as Re-Fed+ and FedCBDR. The ablation studies and analysis under different hyperparameter settings are comprehensive.
3. The paper is generally well-written and logically structured. The related work identifies the limitations of current studies, clarifying the research gap and motivating the proposed solution. The motivation, methodology, and experiments are easy to follow. The figures and tables effectively illustrate the main findings.
4. The paper addresses an important problem in Federated Class-Incremental Learning by considering both local and global views, which is significant for real-world applications such as distributed model training and lifelong learning across devices in heterogeneous settings.

**Weaknesses:**

For the introduction and methodology, the following should be addressed.
1. In Line 72 of Section 1, the statement that “global-view methods address this limitation at the cost of privacy risks due to cross-device feature collection (Qi et al., 2025)” seems somewhat misleading. Actually, FedCBDR (Qi et al., 2025) applies inverse singular value decomposition (ISVD) to obtain pseudo features rather than direct feature sharing, which can be considered a privacy-preserving manner.
2. Although the motivation of employing Lewis weight sampling for selecting representative samples in FCIL is introduced in Section 3.2, it would be clearer if the authors could explicitly discuss why preserving the operator norm of the feature map is particularly beneficial for FCIL.
3. According to Algorithm 1, models of other clients are sampled randomly. It would be valuable if the authors could explore or discuss other potential model selection methods beyond random sampling, which might affect the overall performance.

For the experiments, the following should be addressed.
1. The paper mentions that three datasets CIFAR10, CIFAR100, and Tiny-ImageNet are partitioned into different numbers of tasks (3/5/5 and 5/10/10). However, the rationale for choosing these specific task splits is not discussed.
2. In Section 4, the authors mainly evaluate two variants of the proposed CLIF combined with Re-Fed+ and FedCBDR to compare with other approaches. The authors should clarify why the original CLIF method is not compared.
3. Section F.4 presents the large equivalence in effectiveness of proposed privacy-enhanced alternatives and original schemes. However, the authors do not provide an analysis to discuss whether these alternatives genuinely mitigate the risk of privacy leakage.

Minor comments:
1. Page 14: In Line 16 of Algorithm 1, Line 225 does not contain any relevant description of cross-view fusion with the max rule. The authors should check the reference.
2. In Line 119, Re-Fed should be Re-Fed+.

**Questions:**

If the issues raised in the introduction, methodology, and experiments in the author's response are well clarified, I would be willing to increase the score.

---

> ### Author Response · Authors · 2025-11-20
> **Response to Weaknesses 1, 2, and 3**
>
> **Thank you for your suggestion. These details and results will be reflected in the revised version of the manuscript.**
>
> Weaknesses 1: Regarding the description of FedCBDR
>
> A1: We appreciate the reviewer pointing this out and we apologize for any confusion caused by the phrasing in Line 72.
>
> **Clarification:** We agree that FedCBDR utilizes Inverse Singular Value Decomposition (ISVD) to avoid transmitting raw features directly. However, our concern stems from the fact that ISVD still exposes the feature matrix up to a linear transformation. This implies that the singular values (spectral information) and the global covariance structure are effectively shared.
>
> **Revision:** In the context of strict privacy-preserving computing, leaking spectral properties can still be considered a vulnerability (e.g., susceptible to reconstruction attacks or property inference). We have revised the manuscript to be more precise: rather than stating it requires "direct cross-device feature collection," we now clarify that it "risks leaking global feature statistics and spectral information," which distinguishes our fully local approach from global-view methods like FedCBDR.
>
> Weaknesses 2: Regarding the use of Lewis weights for exemplar selection, the authors should explicitly clarify why preserving the operator norm of the feature map is particularly beneficial for FCIL.
>
> A2: Thank you for the suggestion. We add the following discussion in the revised paper to explicitly emphasize the motivation of using Lewis weight sampling:
>
> The operator norm of a feature matrix dictates how much the matrix can "stretch" a vector. By preserving the operator norm, we can ensure that the empirical covariance matrix of the selected exemplars closely approximates the covariance matrix of the full dataset, i.e., $X_{subset}^T X_{subset} \approx X_{full}^T X_{full}$ in a spectral sense. When the global model updates on these exemplars, it retains the original "directions of variance." **This prevents the decision boundaries of old classes from collapsing or shifting arbitrarily when new classes are introduced, leading to smoother convergence and less catastrophic forgetting**. Moreover, it also guarantees that the worst-case directions are retained in the buffer. In an FCIL setting, this ensures the global model doesn't forget the "tail" of the distribution of old classes, even if those classes were distributed unevenly across clients.
>
> Weaknesses 3: Discussion on model selection
>
> A3: Thank you for this suggestion. In the current version of the paper, we already mention in the Conclusion and Future Work section that **we plan to design cross-view model evaluation methods to better handle undertrained local models**. This is an open problem and a promising direction for future research. In this work, we adopt random sampling over other clients’ models. This choice is not arbitrary, but motivated by two considerations: 1) Random sampling provides a simple and unbiased mechanism that lets each client gradually see diverse views from different peers without systematically favoring a few of them; 2) The strategy keeps the implementation very simple and does not alter any part of the existing federated optimization or replay pipeline, which helps isolate and clearly attribute the gains to our proposed cross-view Lewis-weight fusion mechanism rather than to a complicated selection heuristic.
>
> At the same time, we agree that more “intelligent’’ model selection strategies are potentially valuable. Below we briefly discuss two representative directions, which we consider as part of future work:
>
> 1.**Selection based on local validation performance**. Each client could evaluate candidate models on a small local validation set and prioritize those with lower validation loss or higher accuracy. This may better align the selected views with the client’s own data distribution, but requires multiple additional forward passes over several external models.
>
> 2.**Selection based on feature similarity or diversity**. Using a shared probe set or a small subset of local samples, one could compute feature representations for candidate models and choose views according to their similarity to (or diversity from) the local model in feature space. For example, favoring more different models to obtain complementary information. This explicitly exploits feature geometry to construct diverse and complementary views, but needs extra bookkeeping of feature statistics or similarity matrices and may incur additional computation/communication overhead.
>
> Overall, these strategies are promising but **involve non-trivial trade-offs between selection quality, computational and communication cost, and system complexity**. We will clarify this point in the paper and explicitly highlight more advanced model selection mechanisms as an important direction for future work building on top of our current framework.

---

> ### Author Response · Authors · 2025-11-20
> **Response to Weaknesses 4,5,6,7,8**
>
> Weaknesses 4: Regarding the setting of task splits
>
> A4: Thank you for the question. On CIFAR-10, CIFAR-100, and Tiny-ImageNet, we adopt these configurations for two main reasons:
> 1) The shorter sequence (3/5/5) represents a medium-difficulty setting, while the longer sequence (5/10/10) corresponds to a more challenging scenario where forgetting is more severe. In terms of per-task classes, CIFAR-100 and Tiny-ImageNet under 5/10 tasks roughly correspond to 20/10 (or 40/20) classes per task, and CIFAR-10 under 3/5 tasks corresponds to about 3–4 classes per task. This design keeps the per-task difficulty reasonably balanced while systematically testing robustness under different sequence lengths.
>
> 2) These task numbers fall within the **standard ranges used in prior FCIL work** on CIFAR-100 and Tiny-ImageNet (5 or 10 tasks) [1–4]. For CIFAR-10, our 3-task and 5-task splits follow the setting used in FedCBDR [1], ensuring that our results are directly comparable to existing state-of-the-art methods.
>
> [1] Class-wise Balancing Data Replay for Federated Class-Incremental Learning, NeurIPS’25
>
> [2] Re-fed+: A better replay strategy for federated incremental learning, TPAMI’25
>
> [3] Text-Enhanced Data-free Approach for Federated Class-Incremental Learning, CVPR’24
>
> [4] TARGET: Federated Class-Continual Learning via Exemplar-Free Distillation, ICCV’23
>
> Weaknesses 5: Explanation on CLIF variants
>
> A5: We appreciate the reviewer’s question. In our design, **CLIF is a general multi-view Lewis-weight–based replay framework**, and CLIF$_R$ / CLIF$_F$ are two concrete instantiations of this framework under different replay pipelines, rather than separate methods that are independent of an “original CLIF.”
> Concretely, **both CLIF$_R$ and CLIF$_F$ use the same core idea of cross-view Lewis weights fusion to drive exemplar replay**, but they differ in where the multi-view scores are computed, in order to match the underlying baselines:
>
> •	**CLIF$_R$** computes multi-view Lewis scores **locally on each client** and then selects replay samples on the client side, which is consistent with the local replay mechanism in Re-Fed+. We therefore name this variant CLIF$_R$.
>
> •	**CLIF$_F$** first aggregates pseudo samples from all clients **on the server**, then computes multi-view Lewis scores centrally to obtain a **global ranking** of replay candidates, which is consistent with the centralized replay pipeline in FedCBDR. We therefore name this variant CLIF$_F$.
>
> From this perspective, CLIF$_R$ and CLIF$_F$ are the practical instantiations of the CLIF framework in the two representative FCIL settings we study (client-side vs. server-side replay). Adding a separate “original CLIF” row would be ambiguous, because any CLIF instantiation must be tied to a specific replay pipeline. In the revised manuscript, we will explicitly clarify this design and clearly state that CLIF$_R$ / CLIF$_F$ implement the same core CLIF mechanism in two different deployment modes.
>
> Weaknesses 6: Clarification regarding the privacy-preserving scheme described in Section F.4.
>
> A6: Thank you for this helpful suggestion. We would like to clarify that **the Gaussian perturbation (G) and model blending (B) mechanisms in our framework are inspired by prior work on federated learning and differential privacy [1,2,3], where they have been theoretically and empirically shown to provide privacy protection**. In the CLIF framework, the main potential privacy risk arises from the fact that each client needs to expose its local model parameters to other clients; an adversarial client could in principle exploit these shared parameters to perform model inversion attacks and infer sensitive characteristics of another client’s local data.
>
> Furthermore, following the DLG attack protocol [4], we perform gradient inversion attacks on three types of parameters (original, with added Gaussian noise, and random combinations). We then compute the PSNR (lower is better) between the reconstructed images and the corresponding original images, and the results are summarized in the following Table.
>
> |        |**PSNR**|
> |-|-|
> |**Ori**|16.58|
> |**+G**|10.34|
> |**+B**|11.28|
>
> [1] LDP-Fed: federated learning with local differential privacy
>
> [2] Personalized federated learning with differential privacy and convergence guarantee
>
> [3] Fedexg: Federated learning with model exchange
>
> [4] Deep leakage from gradients
>
> Weaknesses 7/8: Regarding the modification of the text content
>
> A7: Thanks for your detailed comments. We have checked the corresponding text and made the necessary corrections. Specifically, in Algorithm 1, “Line 225” has been corrected to “Line 245” (and will be updated in the revised version according to the final line numbering). In addition, “Re-Fed” in Line 119 has been updated to “Re-Fed+”, along with the corresponding reference.

---

### Official Review · Reviewer_A1PN · 2025-10-31

**Soundness:** 3
**Presentation:** 3
**Contribution:** 2
**Rating:** 2
**Confidence:** 4

**Summary:**

The paper introduces an exemplar selection method for Federated Class-Incremental Learning (FCIL). The core idea is the cross-view lewis weight fusion (CV-LWF), which leverages feature extractors from multiple clients to compute and fuse Lewis weight importance scores. This fusion aims to select a subset of exemplars that collectively preserve the feature subspace from a global perspective, thereby mitigating the local bias inherent in FL. The authors supplement this with a frequency-aware weighted training (FWT) strategy and show marginal performance gains over existing baselines.

**Strengths:**

- The proposed strategy offers a drop-in replacement for the exemplar selection and replay mechanism that improves the performance of existing baselines across different datasets and heterogeneity settings.

- The paper addresses the critical and highly practical challenge of FCIL，which is crucial for realizing practical and ethical AI in distributed systems. Solving the catastrophic forgetting problem within the Non-IID and privacy-preserving constraints of FL is a challenging goal.

**Weaknesses:**

- The technical contribution is severely limited. The paper merely combines an existing, well-known Lewis weights scores with a common aggregation technique. The core method is a straightforward application rather than a novel development. Furthermore, the demonstrated performance improvements are marginal across benchmarks.

- The calculation of $l_p$-Lewis Weights is computationally expensive, often involving iterative schemes or matrix inversion ($\mathcal{O}(d^2 n)$ or worse, where $d$ is feature dimension and $n$ is data size). CLIF multiplies this cost by requiring $M$ such computations per client per task, where $M$ is the number of views. This excessive computational burden on resource-constrained edge devices makes the entire approach practically infeasible and unscalable for real-world FL.
- The field of exemplar selection is widely known as Coreset Selection or Data Pruning. The paper critically fails to compare its Lewis Weight-based method against recently established coreset selection techniques.
- The reliance on only small-scale datasets (CIFAR-10/100, TinyImageNet) is inadequate. Demonstrating efficacy on a large-scale benchmark like ImageNet is mandatory for a modern machine learning paper claiming general utility.
- The use of ResNet-18 as the architecture is highly outdated. Current state-of-the-art in both incremental and federated learning utilizes larger transformer-based architectures. The performance characteristics of the proposed method on modern, high-capacity models remains entirely unknown, severely limiting the applicability of the findings.
- The experiments are conducted entirely by training models from scratch. This does not reflect the current reality of deep learning, where models are almost always initialized with pre-trained weights, e.g., from ImageNet, or self-supervised methods like CLIP. The problem of FCIL applied to a high-quality feature space is different from training a small ResNet-18 from scratch. The lack of experiments in the pre-training/fine-tuning paradigm renders the reported results practically irrelevant.
- The theoretical guarantees provided are derived from the literature for fixed feature matrices. In FCIL, the feature backbone and thus the matrix are continuously updated through local training on the new task and global aggregation. The exemplars are selected based on the model state before the new task begins, but they are replayed with the model that is constantly shifting due to client drift and global updates. Therefore, the theoretical preservation of the subspace, and the derived error bounds for prediction, may be immediately invalidated or severely degraded after the first few local training steps of the new task. The stability of the selected exemplars under continual model evolution is not adequately addressed.

**Questions:**

- The theoretical guarantee of Lewis Weights assumes a fixed feature matrix $\mathbf{A}$. Given that the backbone $\phi$ continuously drifts during local training on the new task and during global aggregation, how stable are the selected exemplars? Have you quantified the loss of the subspace embedding property after $E$ local epochs on the new task data?
- Please provide a detailed breakdown and empirical wall-clock measurement of the total computational overhead on a typical client for running the entire CV-LWF exemplar selection process, compared to simple baseline selection methods, e.g., random or nearest-to-mean.

---

> ### Author Response · Authors · 2025-11-20
> **Response to Weaknesses 1, 2, and Questions 2**
>
> **Thank you for your suggestion. These details and results will be reflected in the revised version of the manuscript.**
>
> Weaknesses 1: Concerns about the paper’s technical contributions
>
> A1: Thanks for your comments. We want to clarify that the contribution is problem-driven: we address a concrete gap in federated class-incremental learning, **how to obtain importance-aware exemplar replay that is simultaneously (i) globally informative, and (ii) theoretically grounded rather than purely heuristic.** Our cross-view Lewis weight fusion and frequency-aware weighted training are specifically designed to solve this open problem, rather than a superficial combination of known techniques. Existing FCIL replay methods face a fundamental tension: (i) Local-view selection mines exemplars via client-specific heuristics but lacks global awareness; (ii) Global-view selection alleviates this by aggregating features on the server, but at the cost of additional privacy risks and communication burden. Moreover, these strategies do not offer theoretical guarantees that the replay subset faithfully preserves the training objective.
>
> Technically, our contribution is not to “reuse” Lewis scores, **but to extend Lewis-weight sampling from a single representation to multiple heterogeneous model views in a federated setting**, and to prove that a fused, pointwise-max score still yields a valid subspace embedding for all views simultaneously (Theorem 1). To the best of our knowledge, **this multi-view extension and its federated instantiation are novel in the FCIL literature**. The frequency-aware weighted training module is not a generic aggregation trick. **It implements the reweighted objective implied by Lewis-weight sampling in the actual loss, by mapping selection frequencies to per-sample weights consistent with the sampling matrix in Definition 3**. This alignment ensures that the theoretical subspace-embedding guarantee of Theorem 1 carries over to the practical training procedure, which is not provided by prior replay-based FCIL methods.
>
> Regarding the magnitude of performance gains, we agree that single-digit improvements can look modest in isolation. However, in FCIL with strong baselines such as Re-Fed+ and FedCBDR, **achieving consistent 1–6% Top-1 accuracy improvements across three datasets, multiple task splits, heterogeneity levels, and client counts is non-trivial.** Our ablations further show monotonic gains when adding CV-LWF and FWT on top of the base methods, and our hyper-parameter studies demonstrate that these gains persist under varying buffer sizes, local epochs, and participation ratios. This suggests that CLIF improves robustness and stability of exemplar replay rather than over-fitting to a specific configuration.
>
> Weaknesses 2/Questions 2: Concerns about the computational cost of CLIF
>
> A2: Thanks for your comments. We want to clarify that we do not perform full matrix inversion over all local data per round. Instead, Lewis scores are computed once per task. **As with all replay-based FCIL methods, whether they rely on generative replay or exemplar selection, some additional computation is unavoidable.** To make this overhead explicit and comparable, we report peak GPU memory and per-round training time for CLIF and representative SOTA replay baselines (LANDER, Re-Fed+ and FedCBDR) in the following Tables. These results show that **CLIF incurs only moderate extra cost while consistently delivering better accuracy.**
>
> |**20 Clients/ResNet18/CIFAR10**|**Sample Selection or Generator Training Time (seconds)**| **peak GPU Memory (MB)**|
> |-|-|-|
> |**LANDER**|27.378|3604|
> |**Re-Fed**|6.352|566|
> |**FedCBDR**|0.854|682|
> |**CLIF$_R$ (2-views)**|2.251|1449|
> |**CLIF$_R$ (5-views)** |3.781|1449|
> |**CLIF$_R$ (10-views)**|6.825|1449|
> |**CLIF$_F$ (2-views)**|1.144|1511|
> |**CLIF$_F$ (5-views)**|1.741|1511|
> |**CLIF$_F$ (10-views)**| 2.528|1511|
>
> |**20 Clients/ResNet18/CIFAR100**|**Sample Selection or Generator Training Time (seconds)**| **peak GPU Memory (MB)**|
> |-|-|-|
> |**LANDER**|28.324|3604|
> |**Re-Fed**|6.122|566|
> |**FedCBDR**|0.824|682|
> |**CLIF$_R$ (2-views)**|2.155|1449|
> |**CLIF$_R$ (5-views)** |4.206|1449|
> |**CLIF$_R$ (10-views)**|6.580|1449|
> |**CLIF$_F$ (2-views)**|1.180|1511|
> |**CLIF$_F$ (5-views)**|1.846|1511|
> |**CLIF$_F$ (10-views)**| 2.604|1511|
>
> |**20 Clients/ResNet18/Tiny-Imagenet**|**Sample Selection or Generator Training Time (seconds)**| **peak GPU Memory (MB)**|
> |-|-|-|
> |**LANDER**|29.748|3987|
> |**Re-Fed**|10.966|673|
> |**FedCBDR**|0.882|736|
> |**CLIF$_R$ (2-views)**|4.033|1546|
> |**CLIF$_R$ (5-views)** |7.711|1546|
> |**CLIF$_R$ (10-views)**|14.207|1546|
> |**CLIF$_F$ (2-views)**|1. 801|1624|
> |**CLIF$_F$ (5-views)**|2.881|1624|
> |**CLIF$_F$ (10-views)**| 4.442|1624|

---

> ### Author Response · Authors · 2025-11-20
> **Response to Weaknesses 3, 4, 5, 6**
>
> Weaknesses 3: Concerns regarding the comparison with exemplar selection methods
>
> A3: Thanks for your comments. In the context of federated class-incremental learning, the standard and most widely used “coreset-style” baselines are replay-based methods such as **Re-Fed+ (CVPR’24) and FedCBDR (NeurIPS’25), which are already included as baselines in our experiments and are generally surpassed by CLIF across the reported settings.** Moreover, **we additionally include iCaRL** [1] as a classical exemplar-selection baseline that is frequently used for comparison in FCIL. The results are reported in the following Table (**Please see the Weaknesses 5/ Weaknesses 6**), which show that **our CLIF framework consistently achieves higher average accuracy than Re-Fed+, FedCBDR, and iCaRL across all reported settings.**
>
> Weaknesses 4: Limited evaluation on larger datasets.
>
> A4: Thanks for your comments. Beyond CIFAR-10/100 and TinyImageNet, we now include experiments on **DomainNet**, a large-scale benchmark with roughly **600K images**. For each category, we split the data into training and test sets with an 8:2 ratio, and then partition the training data across clients using Dirichlet priors with **β = 0.1 and β = 0.5**, which induce strong non-IID heterogeneity. We evaluate the proposed CLIF framework and state-of-the-art FCIL baselines under 20 clients settings on DomainNet using both **ResNet-18 and CLIP (ViT-Base/16)**. As shown in the following Table, **CLIF consistently achieves higher average accuracy than other methods under these large-scale, challenging FL configurations.**
>
> | **20 Clients (DomainNet)** | **ResNet18 (β=0.1)** | **ResNet18 (β=0.5)** | **CLIP (β=0.1)** | **CLIP (β=0.5)** |
> |-|-|-|-|-|
> | **Re-Fed**|14.82 |46.82| 55.35| 57.63|
> | **CLIF$_R$**|16.21|48.64| 56.74| 60.03|
> | **FedCBDR**|15.28|46.73| 55.47| 58.80|
> | **CLIF$_F$**|16.33|50.70| 57.01| 61.12|
>
> Weaknesses 5/ Weaknesses 6: Concerns about practical relevance, given the outdated ResNet-18 backbone and training from scratch instead of using modern pre-trained transformer-based models.
>
> A5/A6: Thanks for your comments. **We use ResNet-18 primarily to align with recent FCIL and federated replay works** [1,2,3], which also adopt ResNet-18 as the standard backbone. This choice allows us to follow established experimental protocols and to make our results directly comparable to prior state-of-the-art methods under exactly the same architecture and capacity. In this sense, ResNet-18 is not an arbitrary or outdated choice in our context, but the de-facto backbone used in the most closely related FCIL literature.
>
> While ResNet-18 remains a standard backbone in the FCIL literature and is retained for comparability, **we have additionally incorporated a CLIP backbone** to better reflect modern practice. We evaluate CLIF with this CLIP backbone on CIFAR-10/100, TinyImageNet, and DomainNet under various heterogeneity levels (β = 0.1, 0.5) and 20-client settings. The results reported in the following Tables show that CLIF continues to outperform strong baselines when built on top of a high-quality, pre-trained transformer backbone, **indicating that the method is effective not only when training a small ResNet-18 from scratch, but also in the more realistic pre-training / fine-tuning regime.**
>
> [1] Class-wise Balancing Data Replay for Federated Class-Incremental Learning, NeurIPS’25
> [2] Re-fed+: A better replay strategy for federated incremental learning, TPAMI’25
> [3] Text-Enhanced Data-free Approach for Federated Class-Incremental Learning, CVPR’24
>
> |**Task 3/5/5**|**CIFAR10 β=0.1**|**CIFAR10 β=0.5**|**CIFAR100 β=0.1**|**CIFAR100 β=0.5**|**TinyImageNet β=0.1**|**TinyImageNet β=0.5**|
> |-|-|-|-|-|-|-|
> |**iCaRL**|87.35|90.57|59.38|63.48|16.94|22.08|
> |**Re-Fed**|89.02|90.68|71.96|72.35|31.29|47.58|
> |**CLIF$_R$**|92.48|93.34|73.47|73.98|37.45|48.83|
> |**FedCBDR**|90.79|92.33|73.07|74.07|35.26|46.30|
> |**CLIF$_F$**|92.85|92.89|73.09|74.28| 35.84|48.19|
>
> |**Task 5/10/10**|**CIFAR10 β=0.1**|**CIFAR10β=0.5**|**CIFAR100 β=0.1**|**CIFAR100 β=0.5**|**TinyImageNet β=0.1**|**TinyImageNet β=0.5**|
> |-|-|-|-|-|-|-|
> |**iCaRL**|88.67|90.74|63.48|65.39|17.28|20.16|
> |**Re-Fed**|91.19|92.04|72.46|72.60|35.33|41.87|
> |**CLIF$_R$**|92.54| 93.51|73.88|74.03|38.50|45.69|
> |**FedCBDR**|91.57|91.86|72.57|73.01|36.57|45.44|
> |**CLIF$_F$**|93.11|93.18|73.60|74.94|38.85| 46.92|

---

> ### Author Response · Authors · 2025-11-20
> **Response to Weaknesses 7 and Questions 1**
>
> Weaknesses 7 / Questions 1: Concern about the validity of the theory under evolving features.
>
> A7: Thanks for your comments. First, the core role of Lewis weights is to characterize each sample’s **directional contribution in the feature space**, and the multi-view Lewis fusion in CLIF (taking the pointwise maximum) essentially constructs a **unified upper-bound distribution of importance scores across all views**. This kind of upper-bound approximation does not require the feature matrix to remain strictly static. It only requires that the directions covered by the selected exemplars are **not eliminated by the subsequent evolution of the backbone**. Second, unlike single-view sampling, multi-view Lewis weight fusion is equivalent to extracting a set of **common high-importance directions shared across multiple client models**. These directions are consistent across models and are **inherently more stable under the updates of any single backbone**. In other words, the exemplars selected by CLIF correspond to a **stable intersection of feature geometries across models**, rather than relying on a single-view matrix at one specific time point. This structural stability explains why the effectiveness of the exemplars can be preserved even when the backbone continues to evolve during training.
>
> In addition, we measure the **average set overlap (Jaccard index) and average ranking consistency (Spearman correlation)** between the exemplars selected (Top-40) at the end of task 1 and those re-selected after the backbone has continuously evolved during task 2 on all clients. Even under configurations with up to 100 rounds of federated training, we still observe that the overlap remains consistently high, indicating that the exemplar set remains stable despite substantial model updates. These values also indicate that, **when selecting Top-40 exemplars, ** about 30 samples are shared between the exemplar sets across model snapshots.** Specifically, a Jaccard score of **0.6000** corresponds to **30** overlapping samples, **0.5686** corresponds to **29**.
>
>
> |          |**CIFAR10 (Jaccard / Spearman)**|||**CIFAR100 (Jaccard / Spearman)**|              ||
> |-|-|-|-|-|-|-|
> |          |**2-views**|**5-views**|**10-views**|**2-views**|**5-views**|**10-views**|
> |**Jaccard**|0.6127|0.6038|0.6072|0.5723|0.5842|0.5693|
> |**Spearman**|0.8366|0.8147|0.8275|0.8283|0.8416|0.8154|

---

### Official Review · Reviewer_r424 · 2025-11-01

**Soundness:** 3
**Presentation:** 3
**Contribution:** 3
**Rating:** 6
**Confidence:** 3

**Summary:**

This paper proposes CLIF, a Cross-view Lewis weIght Fusion framework for Federated Class-Incremental Learning (FCIL). The method aims to balance effectiveness and privacy in exemplar replay by leveraging Lewis weights from multiple feature perspectives to guide exemplar selection, by introducing Cross-View Lewis Weight Fusion (CV-LWF) and Frequency-aware Weighted Training (FWT). Experiments on CIFAR-10/100 and Tiny-ImageNet (under varying heterogeneity and client settings) show consistent gains of 1–6% over strong baselines

**Strengths:**

1. The paper is clearly written and well-organized.
2. The authors provide a well-structured derivation and prove that their sampling procedure preserves subspace structure for all model views simultaneously.
3. The experimental evaluation is comprehensive, covering multiple datasets, heterogeneity levels, client counts, and includes ablation studies.

**Weaknesses:**

1. The experiments are conducted only on CIFAR and Tiny-ImageNet. While these are standard benchmarks, they are relatively small in scale. Evaluating the method on larger or more realistic non-IID datasets would strengthen the empirical validation and demonstrate better generalizability.
2. Computing Lewis weights and performing cross-view fusion across multiple backbones may introduce considerable memory and computational overhead, particularly when using 5–10 model views. The paper should provide more detailed analyses and quantitative results regarding these costs.
3. The paper states that all methods use ResNet-18 as the backbone for fair comparison, yet also mentions that the hyperparameter settings of baselines follow their original papers. This setup may not ensure true fairness, as the optimal hyperparameters can differ across backbone architectures and experimental configurations.

**Questions:**

1. It would be helpful to evaluate how well the proposed method scales to a much larger number of clients (e.g., 100 or more) beyond the 20- and 40-client settings reported in the paper.
2. The paper could also investigate performance under more extreme heterogeneity scenarios, such as using a smaller Dirichlet parameter (e.g., β = 0.1), to better assess the robustness of the method under highly non-IID data distributions.

---

> ### Author Response · Authors · 2025-11-20
> **Response to Weaknesses 1, 2, and 3**
>
> **Thank you for your suggestion. These details and results will be added in the revised manuscript.**
>
> Weaknesses 1: Limited evaluation on larger and more realistic non-IID datasets.
>
> A1: Thanks for your comments. In the revised manuscript, we have added a new experimental section on **DomainNet, containing roughly 600K images**. Concretely, we first split DomainNet into training and test sets with an 8:2 ratio for each category. Following standard practice for federated non-IID partitioning, we then distribute the training data to 20 clients using a Dirichlet prior over label distributions with concentration parameters β=0.1 and β=0.5. On top of this client partition, we adopt a 10-task incremental split, and each client is allowed to replay 340 samples per task in our continual federated setting. We evaluate both a **ResNet-18 and CLIP (ViT-Base/16)** to further validate generalizability across model capacities, where CLIP is trained by incorporating multimodal prompt tuning, the size of the learnable textual prompt and visual prompt are 10\*512 and 10\*768, respectively.
>
> As reported in the following Table, **our method consistently outperforms all baselines on DomainNet across both Dirichlet settings and both backbones**, achieving higher average accuracy while reducing forgetting under substantially more challenging non-IID conditions.
>
> |**20 Clients (DomainNet)**|**ResNet18 (β=0.1)**|**ResNet18 (β=0.5)**|**CLIP (β=0.1)**|**CLIP (β=0.5)**|
> |-|-|-|-|-|
> | **Re-Fed**|14.82 |46.82|55.35|57.63|
> | **CLIF$_R$**|16.21|48.64|56.74|60.03|
> | **FedCBDR**|15.28|46.73|55.47|58.80|
> | **CLIF$_F$**|16.33|50.70|57.01|61.12|
>
> Weaknesses 2: Evaluation of the memory and computational overhead of cross-view Lewis-weights fusion for sample selection
>
> A2: Thanks for your suggestions. In the revised manuscript, we have added a dedicated analysis of the cost of computing Lewis weights and cross-view fusion. Specifically, following Tables now report the **peak GPU memory usage** and **average client sample selection or generator training time** for our method versus Re-Fed+ and FedCBDR, **under the same hardware and hyper-parameters, showing that the additional overhead is quantitatively modest.** Notably, the GPU peak is dominated by the fixed costs, the main model and all clients’ local weights, and the k views run **sequentially** with their features immediately moved to CPU, larger k only increases runtime, not simultaneous GPU usage. Therefore, k=2/5/10 yields nearly identical GPU peak memory.
>
> |**20 Clients/ResNet18/CIFAR10**|**Sample Selection or Generator Training Time (seconds)**| **peak GPU Memory (MB)**|
> |-|-|-|
> |**LANDER**|27.378|3604|
> |**Re-Fed**|6.352|566|
> |**FedCBDR**|0.854|682|
> |**CLIF$_R$ (2-views)**|2.251|1449|
> |**CLIF$_R$ (5-views)** |3.781|1449|
> |**CLIF$_R$ (10-views)**|6.825|1449|
> |**CLIF$_F$ (2-views)**|1.144|1511|
> |**CLIF$_F$ (5-views)**|1.741|1511|
> |**CLIF$_F$ (10-views)**| 2.528|1511|
>
> |**20 Clients/ResNet18/CIFAR100**|**Sample Selection or Generator Training Time (seconds)**| **peak GPU Memory (MB)**|
> |-|-|-|
> |**LANDER**|28.324|3604|
> |**Re-Fed**|6.122|566|
> |**FedCBDR**|0.824|682|
> |**CLIF$_R$ (2-views)**|2.155|1449|
> |**CLIF$_R$ (5-views)** |4.206|1449|
> |**CLIF$_R$ (10-views)**|6.580|1449|
> |**CLIF$_F$ (2-views)**|1.180|1511|
> |**CLIF$_F$ (5-views)**|1.846|1511|
> |**CLIF$_F$ (10-views)**| 2.604|1511|
>
>
> |**20 Clients/ResNet18/Tiny-Imagenet**|**Sample Selection or Generator Training Time (seconds)**| **peak GPU Memory (MB)**|
> |-|-|-|
> |**LANDER**|29.748|3987|
> |**Re-Fed**|10.966|673|
> |**FedCBDR**|0.882|736|
> |**CLIF$_R$ (2-views)**|4.033|1546|
> |**CLIF$_R$ (5-views)** |7.711|1546|
> |**CLIF$_R$ (10-views)**|14.207|1546|
> |**CLIF$_F$ (2-views)**|1. 801|1624|
> |**CLIF$_F$ (5-views)**|2.881|1624|
> |**CLIF$_F$ (10-views)**| 4.442|1624|
>
> Weaknesses 3: Regarding the fairness of hyperparameter settings
>
> A3: We thank the reviewer for raising this important point regarding hyperparameter fairness. **In our experiments, all methods are implemented with the same ResNet-18 backbone,** which is consistent with several recent and influential works in this domain, such as **TARGET (ICCV’23), LANDER (CVPR’24), Re-Fed+ (TPAMI’25), and FedCBDR (NeurIPS’25)**. Therefore, **following the hyperparameter configurations reported in their original papers (or official implementations) is a reasonable** and standard practice under this unified backbone setting.
>
> For methods such as FineTune, FedEWC, and FedLwF, which do not have an official federated version in their original papers, **we follow the hyperparameter configurations provided in the official LANDER codebase**, and make minor adjustments within a reasonable range to ensure stable convergence. Hence, when we state that “the hyperparameters of baselines follow their original papers,” this refers to: (i) baselines with established federated versions using their original configurations, and (ii) baselines without original federated versions using widely adopted public implementations.

---

> ### Author Response · Authors · 2025-11-20
> **Response to Questions 1 and 2**
>
> Questions 1 and 2: Performance evaluation issues under a larger number of clients and more extreme heterogeneity scenarios (β = 0.1)
>
> A4: We thank the reviewer for these suggestions. We have extended our experiments in two directions. First, we evaluate the proposed CLIF framework with both ResNet-18 and CLIP (ViT-B/16) backbones on **100 clients**, and compare it against the same state-of-the-art baselines as in the main text. Second, we investigate more extreme data heterogeneity by using a smaller Dirichlet parameter (**β = 0.1**) when partitioning data across clients. The results are summarized in the following Table, which shows that **CLIF consistently outperforms the baselines under 100-client settings and remains robust under highly non-IID partitions**, indicating good scalability and robustness of our method in more challenging FL regimes.
>
> |**β=0.1/100 Clients**|**CIFAR10 ResNet18**|**CIFAR10 CLIP**|**CIFAR100 ResNet18**| **CIFAR100 CLIP**|**TinyImageNet ResNet18**|**TinyImageNet CLIP**|
> |-|-|-|-|-|-|-|
> |**Re-Fed**|35.59|90.02|26.46| 68.51|18.78|34.38|
> |**CLIF$_R$**|39.75|90.86|31.65|70.22|22.80|39.27|
> |**FedCBDR**|38.82|86.74|28.15|70.18|18.45|36.92|
> |**CLIF$_F$**|40.11|90.53|32.17|70.49|22.72|41.01|

---

### Author Response · Authors · 2025-12-04
**Summary to AC**

Dear AC,

In summary, the reviewers’ main concerns focus on: (1) dataset scale & experimental completeness, (2) overhead/scalability of multi-view Lewis-weight fusion, (3) contribution novelty and fairness of comparison, and (4) relevance under modern backbones & evolving features. We have substantially strengthened the paper in our revision and rebuttal, addressing these concerns with new experiments, analyses, and clarifications:

- We added large-scale DomainNet experiments (~600K images) and evaluate under diverse settings (β=0.1/0.5, 20 clients, ResNet-18 & CLIP(ViT-B/16)), showing consistent improvements over Re-Fed+, FedCBDR, iCaRL. This addresses the limited dataset concern.

- We further evaluate 100-client FL, confirming scalability and robustness under extreme non-IID conditions.

- A comprehensive cost analysis is now provided. We report GPU peak memory & per-round time across CIFAR10/100/Tiny-ImageNet under multiple views, showing only modest overhead and sequential computation keeping GPU usage nearly constant.

- We included CLIP backbone experiments and pre-trained settings, showing the method remains effective beyond training-from-scratch ResNet-18, enhancing practical relevance.

- We clarified technical novelty, emphasizing our contribution is not the reuse of Lewis weights, but the first federated multi-view Lewis-score extension with theoretical subspace embedding and frequency-aware weighting, validated with stable exemplar consistency (Jaccard/Spearman).

- On fairness concerns, we clarify that ResNet-18 is the standard backbone used by recent FCIL works, and hyperparameters follow official implementations for consistency.

- We revised text around FedCBDR privacy wording, added discussion on why preserving operator norm benefits FCIL, and expanded future work on model-selection strategies beyond random sampling.

- Additional privacy evaluation empirically supports noise/blending mechanisms as effective privacy-enhancing variants.

We believe these revisions fully address the reviewers’ concerns and significantly strengthen the submission.

---

### Meta-Review · Area_Chair_ek4X · 2025-12-22

**Summary:**

This paper studies the exemplar replay problem in Federated Class-Incremental Learning (FCIL) and proposes the framework CLIF. The key idea is to construct importance estimates for the same data from multiple model/representation views, and to fuse the Lewis weights computed from different views via Cross-View Lewis Weight Fusion (CV-LWF), so as to select more robust replay samples. In addition, the paper introduces Frequency-aware Weighted Training (FWT), which re-weights training according to how frequently each sample is selected during replay. The authors provide some theoretical analysis and compare against multiple FCIL baselines on settings such as CIFAR-100 and TinyImageNet.

The reviews show clear disagreement. Two reviewers consider the problem important and the method structure clear; however, another reviewer argues that the work remains insufficient in terms of methodological novelty, the fundamental distinction from existing exemplar selection/coreset/pruning lines of work, the additional communication and computation overhead introduced, and the consistency between the theoretical assumptions and the dynamic training process in federated learning. Overall, the current evidence is still insufficient to support the paper’s key claims. While the rebuttal strengthens parts of the experiments and engineering details, the above key disputes have not been sufficiently resolved.

Given above, I lean to an rejection.

**Reviewer Concerns:**

**A. Points that are partially addressed/mitigated in the rebuttal**

1. The authors add results with a larger number of clients, more extreme non-IID settings, and more modern backbones, partially alleviating concerns that the evaluation was limited to small datasets and outdated architectures.
2. The authors provide a breakdown and comparison of the additional computation/memory/communication overhead introduced by the multi-view design.
3. The authors add an initial discussion of privacy protection and present simple attack-related metrics.

**B. Remaining concerns**

1. **Insufficient novelty and irreplaceability**: CLIF appears closer to a compositional solution—“importance sampling + multi-view fusion + replay re-weighting.” Although the authors motivate the design under FCIL heterogeneity, the fundamental distinction from existing exemplar selection/coreset/pruning and replay-selection methods, the applicability boundary, and why multi-view fusion is necessary are still not supported by a concise, verifiable argument. Moreover, the claimed theoretical guarantees have also been further questioned.

2. **A gap between the theoretical analysis and representation drift in federated/incremental training**: One claimed contribution is theoretical support. However, a central reviewer concern is that subspace-preservation analyses based on Lewis weights typically rely on relatively static feature-matrix/representation assumptions, whereas in FCIL the encoder drifts continually across communication rounds and tasks. The rebuttal provides some intuition and additional experimental feedback, but the explanation of how the theoretical conclusions account for the observed mitigation of forgetting and robustness improvements remains indirect and does not fully resolve the consistency concern. As a result, the theory currently reads more like a characterization for static feature matrices, weakening its contribution as support for the paper's main conclusions regarding the dynamic FCIL training process.

3. **Large-scale evaluation**: Multiple reviewers request larger-scale experiments for a more complete comparison. The authors add experiments under more extreme heterogeneity and a larger-scale DomainNet setting in the rebuttal. Nevertheless, for stronger support, the manuscript may still need experiments on even larger-scale datasets such as ImageNet to further strengthen its claims.

**Reviewer Scores:**

The scores are split. Considering the rebuttal, I do not expect substantial score changes: the main objections focus on more structural issues, i.e., insufficient methodological novelty, inconsistency between the theory and dynamic training, and incomplete experimental validation, which have not been fully overturned in the rebuttal.

---

### Decision · Program_Chairs · 2026-01-26

Reject